# AGREE TO DISAGREE: DIVERSITY THROUGH DISAGREEMENT FOR BETTER TRANSFERABILITY

**Matteo Pagliardini**
EPFL

**Martin Jaggi**
EPFL

**François Fleuret**
EPFL

**Sai Praneeth Karimireddy**
EPFL & UC Berkeley

## ABSTRACT

Gradient-based learning algorithms have an implicit *simplicity bias* which in effect can limit the diversity of predictors being sampled by the learning procedure. This behavior can hinder the transferability of trained models by (i) favoring the learning of simpler but spurious features — present in the training data but absent from the test data — and (ii) by only leveraging a small subset of predictive features. Such an effect is especially magnified when the test distribution does not exactly match the train distribution—referred to as the Out of Distribution (OOD) generalization problem. However, given only the training data, it is not always possible to apriori assess if a given feature is spurious or transferable. Instead, we advocate for learning an ensemble of models which capture a diverse set of predictive features. Towards this, we propose a new algorithm D-BAT (Diversity-By-disAgreement Training), which enforces agreement among the models on the training data, but disagreement on the OOD data. We show how D-BAT naturally emerges from the notion of generalized discrepancy, as well as demonstrate in multiple experiments how the proposed method can mitigate shortcut-learning, enhance uncertainty and OOD detection, as well as improve transferability.

## 1 INTRODUCTION

While gradient-based learning algorithms such as Stochastic Gradient Descent (SGD), are nowadays ubiquitous in the training of Deep Neural Networks (DNNs), it is well known that the resulting models are (i) brittle when exposed to small distribution shifts (Beery et al., 2018; Sun et al., 2016; Amodei et al., 2016), (ii) can easily be fooled by small adversarial perturbations (Szegedy et al., 2014), (iii) tend to pick up spurious correlations (McCoy et al., 2019; Oakden-Rayner et al., 2020; Geirhos et al., 2020) — present in the training data but absent from the downstream task — , as well as (iv) fail to provide adequate uncertainty estimates (Kim et al., 2016; van Amersfoort et al., 2020; Liu et al., 2021b). Recently those learning algorithms have been investigated for their implicit bias toward simplicity — known as Simplicity Bias (SB), seen as one of the reasons behind their superior generalization properties (Arpit et al., 2017; Dziugaite & Roy, 2017). While for deep neural networks, simpler decision boundaries are often seen as less likely to overfit, Shah et al. (2020), Pezeshki et al. (2021) demonstrated that the SB can still cause the aforementioned issues. In particular, they show how the SB can be *extreme*, compelling predictors to rely only on the simplest feature available, despite the presence of equally or even more predictive complex features.

Its effect is greatly increased when we consider the more realistic out of distribution (OOD) setting (Ben-Tal et al., 2009), in which the source and target distributions are different, known to be a challenging problem (Sagawa et al., 2020; Krueger et al., 2021). The difference between the two domains can be categorized into either a distribution shift — e.g. a lack of samples in certain parts of the data manifold due to limitations of the data collection pipeline —, or as simply having completely different distributions. In the first case, the SB in its extreme form would increase the chances of learning to rely on spurious features — shortcuts not generalizing to the target distribution. Classic manifestations of this in vision applications are when models learn to rely mostly on textures or backgrounds instead of more complex and likely more generalizable semantic features such as using shapes (Beery et al., 2018; Ilyas et al., 2019; Geirhos et al., 2020). In the second instance, by relying only on the simplest feature, and being invariant to more complex ones, the SB would cause confident predictions (low uncertainty) on completely OOD samples. This even if complex features

---

Correspondence to matteo.pagliardini@epfl.ch and sp.karimireddy@berkeley.edu

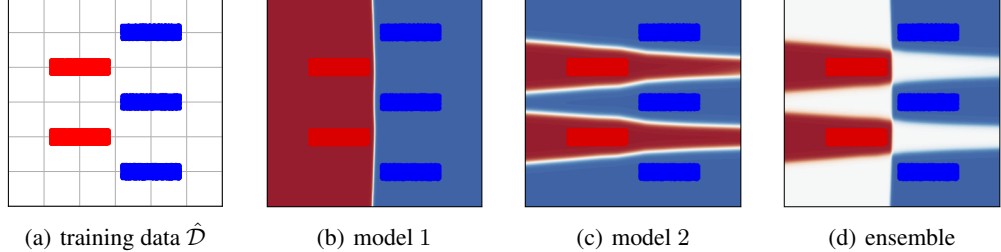

|  (a) training data $\hat{\mathcal{D}}$ | (b) model 1 | (c) model 2 | (d) ensemble |

Figure 1: Example of applying D-BAT on a simple 2D toy example similar to the LMS-5 dataset introduced by Shah et al. (2020). The two classes, red and blue, can easily be separated by a vertical boundary decision. Other ways to separate the two classes — with horizontal lines for instance — are more complex., i.e. they require more hyperplanes. The simplicity bias will push models to systematically learn the simpler feature, as in the second column (b). Using D-BAT, we are able to learn the model in column (c), relying on a more complex boundary decision, effectively overcoming the simplicity bias. The ensemble $h_{ens}(x) = h_1(x) + h_2(x)$, in column (d), outputs a flat distribution at points where the two models disagree, effectively maximizing the uncertainty at those points. In this experiments the samples from $\mathcal{D}_{ood}$ were obtained through computing adversarial perturbations, see App. D.2 for more details.

are contradicting simpler ones. Which brings us to our goal of deriving a method which can (i) learn more transferable features, better suited to generalize despite distribution shifts, and (ii) provides accurate uncertainty estimates also for OOD samples.

We aim to achieve those two objectives through learning an ensemble of diverse predictors $(h_1, \ldots, h_K)$, with $h : \mathcal{X} \to \mathcal{Y}$, and $K$ being the ensemble size. Suppose that our training data is drawn from the distribution $\mathcal{D}$, and $\mathcal{D}_{ood}$ is the distribution of OOD data on which we will be tested. Importantly, $\mathcal{D}$ and $\mathcal{D}_{ood}$ may have non-overlapping support, and $\mathcal{D}_{ood}$ is not known during training. Our proposed method, D-BAT (Diversity-By-disAgreement Training), relies on the following idea:

*Diverse hypotheses should agree on the source distribution $\mathcal{D}$ while disagreeing on the OOD distribution $\mathcal{D}_{ood}$.*

Intuitively, a set of hypotheses should agree on what is known i.e. on $\mathcal{D}$, while formulating different interpretations of what is *not* known, i.e. on $\mathcal{D}_{ood}$. Even if each *individual predictor* might be wrongly confident on OOD samples, while predicting different outcomes — the resulting uncertainty of the *ensemble* on those samples will be increased. Disagreement on $\mathcal{D}_{ood}$ can itself be enough to promote learning diverse representations of instances of $\mathcal{D}$. In the context of object detection, if one model $h_1$ is relying on textures only, this model will generate predictions on $\mathcal{D}_{ood}$ based on textures, when enforcing disagreement on $\mathcal{D}_{ood}$, a second model $h_2$ would be discouraged to use textures in order to disagree with $h_1$ — and consequently look for a different hypothesis to classify instances of $\mathcal{D}$ e.g. using shapes. This process is illustrated in Fig. 2. A 2D direct application of our algorithm can be seen in Fig. 1. Once trained, the ensemble can either be used by forming a weighted average of the probability distribution from each hypothesis, or—if given some labeled data from the downstream task—by selecting one particular hypothesis.

**Contributions.** Our results can be summarized as:

- We introduce D-BAT, a simple yet efficient novel diversity-inducing regularizer which enables training ensembles of diverse predictors.

- We provide a proof, in a simplified setting, that D-BAT promotes diversity, encouraging the models to utilize different predictive features.

- We show on several datasets of varying complexity how the induced diversity can help to (i) tackle shortcut learning, and (ii) improve uncertainty estimation and transferability.

## 2 RELATED WORK

**Diversity in ensembles.** It is intuitive that in order to gain from ensembling several predictors $h_1, ..., h_K$, those should be diverse. The bias-variance-covariance decomposition (Ueda & Nakano, 1996), which generalizes the bias variance decomposition to ensembles, shows how the error decreases with the covariance of the members of the ensemble. Despite its importance, there is still no well accepted definition and understanding of diversity, and it is often derived from prediction errors of members of the ensemble (Zhou, 2012). This creates a conflict between trying to increase accuracy of individual predictors $h$, and trying to increase diversity. In this view, creating a good ensemble is seen as striking a good balance between individual performance and diversity. To promote diversity in ensembles, a classic approach is to add stochasticity into the training by using different subsets of the training data for each predictor (Breiman, 1996), or using different data augmentation methods (Stickland & Murray, 2020). Another approach is to add orthogonality constrains on the predictor's gradient (Ross et al., 2020; Kariyappa & Qureshi, 2019). Recently, the information bottleneck (Tishby et al., 2000) has been used to promote ensemble diversity (Ramé & Cord, 2021; Sinha et al., 2021). Unlike the aforementioned methods, D-BAT can be trained on the full dataset, it importantly does not set constrains on the output of in-distribution samples, but on a separate OOD distribution. Moreover, as opposed to Sinha et al. (2021), our individual predictors do not share the same encoder.

**Simplicity bias.** While the simplicity bias, by promoting simpler decision boundary, can act as an implicit regularizer and improves generalization (Arpit et al., 2017; Gunasekar et al., 2018), it is also contributing to the brittleness of gradient-based machine-leaning (Shah et al., 2020). Recently Teney et al. (2021) proposed to evade the simplicity bias by adding gradient orthogonality constrains, not at the output level, but at an intermediary hidden representation obtained after a shared and fixed encoder. While their results are promising, the reliance on a pre-trained encoder limits the type of features that can be used to the set of features extracted by the encoder, especially, if a feature was already discarded by the encoder due to SB, it is effectively lost. In contrast, our method is not relying on a pre-trained encoder, also comparatively require a very small ensemble size to counter the simplicity bias. A more detailed comparison with D-BAT is provided in App F.1.

**Shortcut learning.** The failures of DNNs across application domains due to shortcut learning have been documented extensively in (Geirhos et al., 2020). They introduce a taxonomy of predictors distinguishing between (i) predictors which can be learnt from the training algorithms (ii) predictors performing well on in-distribution training data, (iii) predictors performing well on in-distribution test data, and finally (iv) predictors performing well on in-distribution and OOD test data. The last category being the intended solutions. In our experiments, by learning diverse predictors, D-BAT increases the chance of finding one solution generalizing to both in and out of distribution test data, see § 4.1 for more details.

**OOD generalization.** Generalizing to distributions not seen during training is accomplished by two approaches: robust training, and invariant learning. In the former, the test distribution is assumed to be within a set of known plausible distributions (say $\mathcal{U}$). Then, robust training minimizes the loss over the worst possible distribution in $\mathcal{U}$ (Ben-Tal et al., 2009). Numerous approaches exist to defining the set $\mathcal{U}$ - see survey by (Rahimian & Mehrotra, 2019). Most recently, Sagawa et al. (2020) model the set of plausible domains as the convex hull over predefined subgroups of datapoints and Krueger et al. (2021) extend this by taking affine combinations beyond the convex hull. Our approach also borrows from this philosophy - when we do not know the labels of the OOD data, we assume the worst case and try predict as diverse labels as possible. This is similar to the notion of discrepancy introduced in domain adaptation theory (Mansour et al., 2009; Cortes & Mohri, 2011; Cortes et al., 2019). A different line of work defines a set of environments and asks that our outputs be 'invariant' (i.e. indistinguishable) among the different environments (Bengio et al., 2013; Arjovsky et al., 2019; Koyama & Yamaguchi, 2020). When only a single training environment is present, like in our setting, this is akin to adversarial domain adaptation. Here, the data of one domain is modified to be indistinguishable to the other (Ganin et al., 2016; Long et al., 2017). However, this approach is fundamentally limited. E.g. in Fig. 2 a model which classifies both the crane and the porcupine as a crane is invariant, but incorrect. Furthermore, it is worth noting that prior work in OOD generalization are often considering datasets where the spurious feature is *not* fully predictive in the training distribution (Zhang et al., 2021; Saito et al., 2017; 2018; Nam et al., 2020; Liu et al., 2021a), and fail in our challenging settings of § 4.1 (see App. F for more in-depth comparisons). Lastly, parallel to our work, Lee et al. (2022) adopt a similar approach and improve OOD generalization by

Figure 2: Illustration of how D-BAT can promote learning diverse features. Consider the task of classifying bird pictures among several classes. The red color represents the attention of a first model $h_1$. This model learnt to use some simple yet discriminative feature to recognise an African Crowned Crane on the left. Now suppose we use the top image $\mathcal{D}_{\text{ood}}$ on which the models must disagree. $h_2$ cannot again use the same feature as $h_1$ since then it will not disagree on $\mathcal{D}_{\text{ood}}$. Instead, $h_2$ would look for other distinctive features of the crane which are not present on the right e.g. using its beak and red throat pouch.

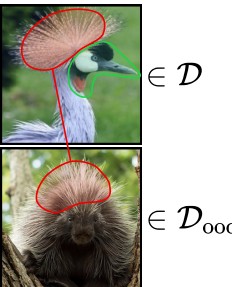

minimizing the mutual information on unlabeled target data between pairs of predictors. However, their work does not investigate uncertainty estimation and is not motivated by domain adaptation theory as ours is (Mansour et al., 2009), see App. F.7 for a more in-depth comparison.

**Uncertainty estimation.** DNNs are notoriously unable to provide reliable confidence estimates, which is impeding the progress of the field in safety critical domains (Begoli et al., 2019), as well as hurting models interpretability (Kim et al., 2016). To improve the confidence estimates of DNNs, Gal & Ghahramani (2016) propose to use dropout at inference time, a method referred to as MC-Dropout. Other popular methods used for uncertainty estimation are Bayesian Neural Networks (BNNs) (Hernández-Lobato & Adams, 2015) and Gaussian Processes (Rasmussen & Williams, 2005). All those methods but gaussian processes, were recently shown to fail to adequately provide high uncertainty estimates on OOD samples *away* from the boundary decision (van Amersfoort et al., 2020; Liu et al., 2021b). We show in our experiments how D-BAT can help to associate high uncertainty to those samples by maximizing the disagreement outside of $\mathcal{D}$ (see § 4.2, as well as Fig.1).

## 3  DIVERSITY THROUGH DISAGREEMENT

### 3.1  MOTIVATING D-BAT

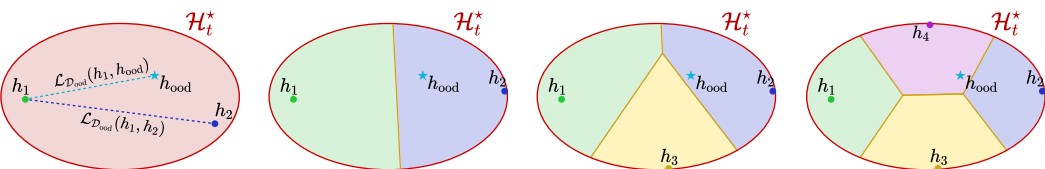

Figure 3: If $h_1$ is computed by minimizing the training loss on $\mathcal{D}$, its loss on the OOD task $\mathcal{D}_{\text{ood}}$ may be very large i.e. $h_1$ may be very far from the optimal OOD model $h_{\text{ood}}$ as measured by $\mathcal{L}_{\mathcal{D}_{\text{ood}}}(h_1, h_{\text{ood}})$ (left). To mitigate this, we propose to learn a diverse ensemble $\{h_1, \ldots, h_4\}$ which is maximally 'spread-out' (with distance measured using $\mathcal{L}_{\mathcal{D}_{\text{ood}}}(\cdot, \cdot)$) and cover the entire space of possible solutions $\mathcal{H}_t^\star$. This minimizes the distance between the unknown $h_{\text{ood}}$ and our learned ensemble, ensuring we learn transferable features with good performance on $\mathcal{D}_{\text{ood}}$.

We will first define some notation and explain why standard training fails for OOD generalization. Then, we introduce the concept of discrepancy which will motivate our D-BAT algorithm.

**Setup.**  Let us formally define the OOD problem. $\mathcal{X}$ is the input space, $\mathcal{Y}$ the output space, we define a domain as a pair of a distribution over $\mathcal{X}$ and a labeling function $h : \mathcal{X} \to \mathcal{Y}$. Given any distribution $\mathcal{D}$ over $\mathcal{X}$, given two labeling functions $h_1$ and $h_2$, given a loss function $L : \mathcal{Y} \times \mathcal{Y} \to \mathbb{R}_+$, we define the expected loss as the expectation: $\mathcal{L}_{\mathcal{D}}(h_1, h_2) = \mathbb{E}_{x \sim \mathcal{D}}[L(h_1(x), h_2(x))]$.

Now, suppose that the training data is drawn from the distribution $(\mathcal{D}_t, h_t)$, but we will be tested on a different distribution $(\mathcal{D}_{\text{ood}}, h_{\text{ood}})$. While the labelling function $h_{\text{ood}}$ is unknown, we assume that we have access to unlabelled samples from $\mathcal{D}_{\text{ood}}$.

Finally, let $\mathcal{H}$ be the set of all labelling functions i.e. the set of all possible prediction models. And further define $\mathcal{H}_t^\star$ and $\mathcal{H}_{\text{ood}}^\star$ to be the optimal labelling functions on the train and the OOD domains:

$$\mathcal{H}_t^\star := \arg\min_{h \in \mathcal{H}} \mathcal{L}_{\mathcal{D}_t}(h, h_t), \quad \mathcal{H}_{\text{ood}}^\star := \arg\min_{h \in \mathcal{H}} \mathcal{L}_{\mathcal{D}_{\text{ood}}}(h, h_{\text{ood}}).$$

We assume that there exists an ideal transferable function $h^\star \in \mathcal{H}_t^\star \cap \mathcal{H}_{\text{ood}}^\star$. This assumption captures the reality that the training task and the OOD testing task are closely related to each other. Otherwise, we would not expect any OOD generalization.

**Beyond standard training.** Just using the training data, standard training would train a model $h_{\text{ERM}} \in \mathcal{H}_t^\star$. However, as we discussed in the introduction, if we use gradient descent to find the ERM solution, then $h_{\text{ERM}}$ will likely be the simplest model i.e. it will likely pick up spurious correlations in $\mathcal{D}_t$ which are not present in $\mathcal{D}_{\text{ood}}$. Thus, the error on OOD data might be very high

$$\mathcal{L}_{\mathcal{D}_{\text{ood}}}(h_{\text{ERM}}, h_{\text{ood}}) \leq \max_{h \in \mathcal{H}_t^\star} \mathcal{L}_{\mathcal{D}_{\text{ood}}}(h, h_{\text{ood}}).$$

Instead, we would ideally like to *minimize* the right hand side in order to find $h^\star$. The main difficulty is that we do not have access to the OOD labels $h_{\text{ood}}$. So we can instead use the following proxy:

$$\mathcal{L}_{\mathcal{D}_{\text{ood}}}(h_1, h_{\text{ood}}) = \max_{h_2 \in \mathcal{H}_t^\star \cap \mathcal{H}_{\text{ood}}^\star} \mathcal{L}_{\mathcal{D}_{\text{ood}}}(h_1, h_2) \leq \max_{h_2 \in \mathcal{H}_t^\star} \mathcal{L}_{\mathcal{D}_{\text{ood}}}(h_1, h_2)$$

In the above we used the two following facts, (i) that $\forall h_2 \in \mathcal{H}_{\text{ood}}^\star$, $\mathcal{L}_{\mathcal{D}_{\text{ood}}}(h_1, h_{\text{ood}}) = \mathcal{L}_{\mathcal{D}_{\text{ood}}}(h_1, h_2)$, as well as (ii) that $\mathcal{H}_t^\star \cap \mathcal{H}_{\text{ood}}^\star$ is non-empty. Recall that $\mathcal{H}_t^\star = \arg\min_{h \in \mathcal{H}} \mathcal{L}_{\mathcal{D}_t}(h, h_{\text{t}})$. So this means — in order to minimize the upper bound — we want to pick $h_2$ to minimize risk on our training data (i.e. belong to $\mathcal{H}_t^\star$), but otherwise maximally disagree with $h_1$ on the OOD data. That way we minimize the worst case expected loss: $\min_{h \in \{h_1, h_2\}} \max_{h' \in \mathcal{H}_t^\star} \mathcal{L}_{\mathcal{D}_{\text{ood}}}(h, h')$ — this process is illustrated in Fig. 3. The latter is closely related to the concept of discrepancy in domain-adaption (Mansour et al., 2009; Cortes et al., 2019). However, the main difference between the definitions is that we restrict the maximum to the set of $\mathcal{H}_t^\star$, whereas the standard notions use an unrestricted maximum. Thus, our version is tighter when the train and OOD tasks are closely related.

**Deriving D-BAT.** We make two final changes to the discrepancy term above to derive D-BAT. First, if $\mathcal{L}_{\mathcal{D}}(h_1, h_2)$ is a loss function which quantifies dis-agreement, then suppose we have another loss function $\mathcal{A}_{\mathcal{D}}(h_1, h_2)$ which quantifies agreement. Then, we can minimize agreement instead of maximizing dis-agreement

$$\arg\min_{h_2 \in \mathcal{H}_t^\star} \mathcal{A}_{\mathcal{D}}(h_1, h_2) = \arg\max_{h_2 \in \mathcal{H}_t^\star} \mathcal{L}_{\mathcal{D}}(h_1, h_2).$$

Secondly, we relax the constrained formulation $h_2 \in \mathcal{H}_t^\star$ by adding a penalty term with weight $\alpha$ as

$$h_{\text{D-BAT}} \in \min_{h_2 \in \mathcal{H}} \underbrace{\mathcal{L}_{\mathcal{D}_t}(h_2, h_t)}_{\text{fit train data}} + \alpha \underbrace{\mathcal{A}_{\mathcal{D}_{\text{ood}}}(h_1, h_2)}_{\text{disagree on OOD}}.$$

The above is the core of our D-BAT procedure - given a first model $h_1$, we train a second model $h_2$ to fit the training data $\mathcal{D}$ while disagreeing with $h_1$ on $\mathcal{D}_{\text{ood}}$. Thus, we have

$$\mathcal{L}_{\mathcal{D}_{\text{ood}}}(h_1, h_{\text{ood}}) \leq \max_{h_2 \in \mathcal{H}_t^\star} \mathcal{L}_{\mathcal{D}_{\text{ood}}}(h_1, h_2) \approx \mathcal{L}_{\mathcal{D}_{\text{ood}}}(h_1, h_{\text{D-BAT}}),$$

implying that D-BAT gives us a good proxy for the unknown OOD loss, and can be used for uncertainty estimation. Following a similar argument for $h_1$, we arrive the following training procedure:

$$\min_{h_1, h_2} \tfrac{1}{2}(\mathcal{L}_{\mathcal{D}_t}(h_1, h_t) + \mathcal{L}_{\mathcal{D}_t}(h_2, h_t)) + \alpha \mathcal{A}_{\mathcal{D}_{\text{ood}}}(h_1, h_2).$$

However, we found the training dynamics for simultaneously learning $h_1$ and $h_2$ to be unstable. Hence, we propose a sequential variant which we describe next.

## 3.2 ALGORITHM DESCRIPTION

**Binary classification formulation.** Concretely given a binary classification task, with $\mathcal{Y} = \{0, 1\}$, we train two models sequentially. The training of the first model $h_1$ is done in a classical way, minimizing its empirical classification loss $\mathcal{L}(h_1(\boldsymbol{x}), y)$ over samples $(\boldsymbol{x}, y)$ from $\hat{\mathcal{D}}$. Once $h_1$ trained, we train the second model $h_2$ adding a term $\mathcal{A}_{\tilde{\boldsymbol{x}}}(h_1, h_2)$ representing the agreement on samples $\tilde{\boldsymbol{x}}$ of $\hat{\mathcal{D}}_{\text{ood}}$, with some weight $\alpha \geq 0$:

$$h_2^\star \in \arg\min_{h_2 \in \mathcal{H}} \tfrac{1}{N} \Big( \sum_{(\boldsymbol{x}, y) \in \hat{\mathcal{D}}} \mathcal{L}(h_2(\boldsymbol{x}), y) + \alpha \sum_{\tilde{\boldsymbol{x}} \in \hat{\mathcal{D}}_{\text{ood}}} \mathcal{A}_{\tilde{\boldsymbol{x}}}(h_1, h_2) \Big)$$

Given $p_{h,\boldsymbol{x}}^{(y)}$ the probability of class $y$ predicted by $h$ given $\boldsymbol{x}$, the agreement $\mathcal{A}_{\tilde{\boldsymbol{x}}}(h_1, h_2)$ is defined as:

$$\mathcal{A}_{\tilde{\boldsymbol{x}}}(h_1, h_2) = -\log\left(p_{h_1,\tilde{\boldsymbol{x}}}^{(0)} \cdot p_{h_2,\tilde{\boldsymbol{x}}}^{(1)} + p_{h_1,\tilde{\boldsymbol{x}}}^{(1)} \cdot p_{h_2,\tilde{\boldsymbol{x}}}^{(0)}\right) \tag{AG}$$

In the above formula, the term inside the $\log$ can be derived from the expected loss when $L$ is the 01-loss and $h_1$, $h_2$ independent. See App. B for more details.

**Multi-class classification formulation.** The previous formulation requires a distribution over two labels in order to compute the agreement term (AG). We extend the agreement term $\mathcal{A}(h_1, h_2, \tilde{\boldsymbol{x}})$ to the multi-class setting by binarizing the softmax distributions $h_1(\tilde{\boldsymbol{x}})$ and $h_2(\tilde{\boldsymbol{x}})$. A simple way to do this is to take as positive class the predicted class of $h_1$: $\tilde{y} = \operatorname{argmax}(h_1(\tilde{\boldsymbol{x}}))$ with associated probability $p_{h_1,\tilde{\boldsymbol{x}}}^{(\tilde{y})}$, while grouping the remaining complementary class probabilities in a negative class $\neg\tilde{y}$. We would then have $p_{h_1,\tilde{\boldsymbol{x}}}^{(\neg\tilde{y})} = 1 - p_{h_1,\tilde{\boldsymbol{x}}}^{(\tilde{y})}$. We can then use the same bins to binarize the softmax distribution of the second model $h_2(\tilde{x})$. Another similarly sound approach would be to do the opposite and use the predicted class of $h_2$ instead of $h_1$. In our experiments both approaches performed well. In Alg.2 we show the second approach, which is a bit more computationally efficient in the case of ensembles of more than 2 predictors, as the binarization bins are built only once, instead of building them for each pair $(h_i, h_m)$ for $0 \le i < m$.

### 3.3 LEARNING DIVERSE FEATURES

It is possible, under some simplifying assumptions to rigorously prove that minimizing $\mathcal{L}_{\text{D-BAT}}$ results in learning predictors which use diverse features. We introduce the following theorem:

**Theorem 3.1** (D-BAT favors diversity). *Given a joint source distribution $\mathcal{D}$ of triplets of random variables $(C, S, Y)$ taking values in $\{0, 1\}^3$. Assuming $\mathcal{D}$ has the following PMF: $\mathbb{P}_{\mathcal{D}}(C = c, S = s, Y = y) = 1/2$ if $c = s = y$, and $0$ otherwise, which intuitively corresponds to experiments § 4.1 in which two features (e.g. color and shape) are equally predictive of the label $y$. Assuming a first model learnt the posterior distribution $\mathbb{P}_1(Y = 1 \mid C = c, S = s) = c$, meaning that it is invariant to feature $s$. Given a distribution $\mathcal{D}_{ood}$ uniform over $\{0, 1\}^3$ outside of the support of $\mathcal{D}$, the posterior solving the D-BAT objective will be $\mathbb{P}_2(Y = 1 \mid C = c, S = s) = s$, invariant to feature $c$.*

The proof is provided in App. C. It crucially relies on the fact that $\mathcal{D}_{\text{ood}}$ has positive weight on data points which only contain the alternative feature $s$, or only contain the feature $c$. Thus, as long as $\mathcal{D}_{\text{ood}}$ is supported on a diverse enough dataset with features present in different combinations , we can expect D-BAT to learn models which utilize a variety of such features.

## 4 EXPERIMENTS

We conduct two main types of experiments, (i) we evaluate how D-BAT can mitigate shortcut learning, bypassing simplicity bias, and generalize to OOD distributions, and (ii) we test the uncertainty estimation and OOD detection capabilities of D-BAT models.

### 4.1 OOD GENERALIZATION AND AVOIDING SHORTCUTS

We estimate our method's ability to avoid spurious correlation and learn more transferable features on 6 different datasets. In this setup, we use a labelled training data $\mathcal{D}$ which might have a lot of highly correlated spurious features, and an unlabelled perturbation dataset $\mathcal{D}_{ood}$. We then test the performance on the learnt model on a test dataset. This test dataset may be drawn from the same distribution as $\mathcal{D}_{ood}$ (which tests how well D-BAT avoids spurious features), as well as from a completely different distribution from $\mathcal{D}_{ood}$ (which tests if D-BAT generalizes to new domains). We compare D-BAT against ERM, both when used to obtain a single model or an ensemble.

Our results are summarized in Tab. 1. For each dataset, we report both the best-model accuracy and — when applicable — the best-ensemble accuracy. All experiments in Tab. 1 are with an ensemble of size 2. Among the two models of the ensemble, the best model is selected according to its validation accuracy. We show results for a larger ensemble size of 5 in Fig. 4. Finally in Fig. 4 C (right) we compare the performance of D-BAT against numerous other baseline methods. See Appendix D for additional details on the setup as well as numerous other results.

Table 1: Test accuracies on the six datasets described in § 4.1. For each dataset, we compare single model and ensemble test accuracies for D-BAT and ERM. In the left column we consider the scenario where $\mathcal{D}_{ood}$ is also our test distribution (we can imagine we have access to unlabeled data from the test distribution). In the right column we consider $\mathcal{D}_{ood}$ and our test distribution to be different, e.g. belonging to different domains. see § 4.1 for more details and a summary of our findings. In bold are the best scores along with any score within standard deviation reach. For datasets with completely spurious correlations, as we know ERM models would fail to learn anything generalizable, we are not interested in using them in a ensemble, hence the missing values for those datasets.

| | $\mathcal{D}_{ood} =$ test data (unlabelled) | | | | $\mathcal{D}_{ood} \neq$ test data | | | |
| | Single Model | | Ensemble | | Single Model | | Ensemble | |
| Dataset $\mathcal{D}$ | ERM | D-BAT | ERM | D-BAT | ERM | D-BAT | ERM | D-BAT |
| --- | --- | --- | --- | --- | --- | --- | --- | --- |
| C-MNIST | $12.3 \pm 0.7$ | $\mathbf{90.2 \pm 3.7}$ | - | - | $27.1 \pm 2.8$ | $\mathbf{90.1 \pm 1.9}$ | - | - |
| M/F-D | $52.9 \pm 0.1$ | $\mathbf{94.8 \pm 0.3}$ | - | - | $52.9 \pm 0.1$ | $\mathbf{89.0 \pm 0.6}$ | - | - |
| M/C-D | $50.0 \pm 0.0$ | $\mathbf{73.3 \pm 1.2}$ | - | - | $50.0 \pm 0.0$ | $\mathbf{58.0 \pm 0.6}$ | - | - |
| Waterbirds | $86.0 \pm 0.5$ | $\mathbf{88.7 \pm 0.2}$ | $85.8 \pm 0.4$ | $\mathbf{87.5 \pm 0.0}$ | - | - | - | - |
| Office-Home | $50.4 \pm 1.0$ | $\mathbf{51.1 \pm 0.7}$ | $52.0 \pm 0.5$ | $\mathbf{52.7 \pm 0.2}$ | $\mathbf{51.7 \pm 0.6}$ | $\mathbf{51.7 \pm 0.3}$ | $53.9 \pm 0.4$ | $\mathbf{54.5 \pm 0.5}$ |
| Camelyon17 | $80.3 \pm 0.4$ | $\mathbf{93.1 \pm 0.3}$ | $80.9 \pm 1.5$ | $\mathbf{91.9 \pm 0.4}$ | $80.3 \pm 0.4$ | $\mathbf{88.8 \pm 1.4}$ | $80.9 \pm 1.5$ | $\mathbf{85.9 \pm 0.9}$ |

**Training data ($\mathcal{D}$).**   We consider two kinds of training data: synthetic datasets with completely spurious correlation, and more real world datasets where do not have any control and naturally may have some spurious features. We use the former to have a controlled setup, and the latter to judge our performance in the real world.

*Datasets with completely spurious correlation*: To know whether we learn a shortcut, and estimate our method's ability to overcome the SB, we design three datasets of varying complexity with known shortcut in a similar fashion as Teney et al. (2021). The Colored-MNIST, or C-MNIST for short, consists of MNIST (Lecun & Cortes, 1998) images for which the color and the shape of the digits are equally predictive, i.e. all the 1 are pink, all the 5 are orange, etc. The color being simpler to learn than the shape, the simplicity bias will result in models trained on this dataset to rely solely on the color information while being invariant to the shape information. This dataset is a multiclass dataset with 10 classes. The test distribution consists of images where the label is carried by the shape of the digit and the color is random. Following a similar idea, we build the M/F-Dominoes (M/F-D) dataset by concatenating MNIST images of 0s and 1s with Fashion-MNIST (Xiao et al., 2017) images of coats and dresses. The source distribution consists in images where the MNIST and F-MNIST parts are equally predicitve of the label. In the test distribution, the label is carried by the F-MNIST part and the MNIST part is a 0 or 1 MNIST image picked at random. The M/C-Dominoes (M/C-D) dataset is built in the same way concatenating MNIST digits 0s and 1s with CIFAR-10 (Krizhevsky, 2009) images of cars and trucks. See App. E to see samples from those datasets.

*Natural datasets*: To test our method in this more general case we run further experiments on three well-known domain adaptation datasets. We use the Waterbirds (Sagawa et al., 2020) and Camelyon17 (Bandi et al., 2018) datasets from the WILDS collection (Koh et al., 2021). Camelyon17 is an image dataset for cancer detection, where different hospital each provide a unique data part. For those two binary classification datasets, the test distributions are taken to be the pre-defined test splits. We also use the Office-Home dataset from Venkateswara et al. (2017), which consists of images of 65 item categories across 4 domains: Art, Product, Clipart, and Real-World. In our experiments we merge the Product and Clipart domains to use as training, and test on the Real-World domain.

**Perturbation data ($\mathcal{D}_{ood}$).**   As mentioned previously, we consider two scenarios in which the test distribution is (i) drawn from the same distribution as $\mathcal{D}_{ood}$, or (ii) drawn from a completely different distribution. In practice, in the later case, we keep the test distribution unchanged and modify $\mathcal{D}_{ood}$. For the C-MNIST, we remove digits 5 to 9 from the training and test distributions and build $\mathcal{D}_{ood}$ based on those digits associated with random colors. For M/F-D and M/C-D datasets, we build $\mathcal{D}_{ood}$ by concatenating MNIST images of 0 and 1 with F-MNNIST, — respectively CIFAR-10 — categories which are not used in the training distribution (i.e. anything but coats and dresses, resp. trucks and cars), samples from those distributions are in App. E. For the Camelyon17 medical imaging dataset, we use unlabeled validation data instead of unlabeled test data, both coming from different hospitals. For the Office-Home dataset, we use the left-out Art domain as $\mathcal{D}_{ood}$.

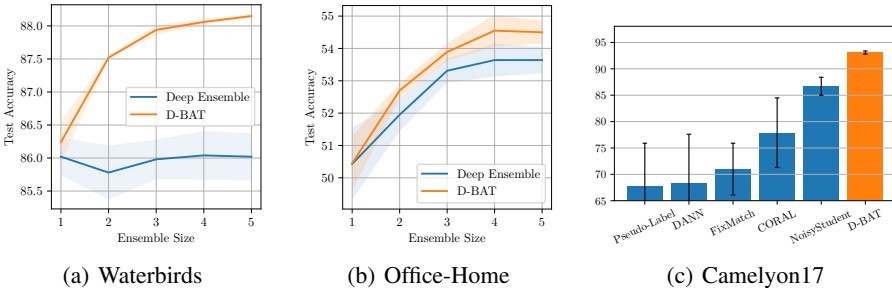

| (a) Waterbirds | (b) Office-Home | (c) Camelyon17 |

Figure 4: All results are in the "$\mathcal{D}_{\text{ood}}$ = test data" setting. **(a)** and **(b)**: Test accuracies as a function of the ensemble size for both D-BAT and Deep Ensembles (ERM ensembles). We observe a significant advantage of D-BAT on both the Waterbirds and the Office-Home datasets. The difference is especially visible on the Waterbirds dataset, which has a stronger spurious correlation. Results have been obtained averaging over 3 seeds for the Waterbirds dataset and 6 seeds for the Office-Home dataset. **(c)**: Comparison of D-BAT with several other methods on the Camelyon17, results except D-BAT are taken from Sagawa et al. (2022).

**Results and discussion.**

- **D-BAT can tackle extreme spurious correlations.** This is unlike prior methods from domain adaptation (Zhang et al., 2021; Saito et al., 2017; 2018; Nam et al., 2020; Liu et al., 2021a) which all fail when the spurious feature is completely correlated with the label, see App. F for an extended discussion and comparison in which we show those methods cannot improve upon ERM in that scenario. First we look at results without D-BAT for the C-MNIST, M/F-D and M/C-D datasets in Tab. 1. Looking at the ERM column, we observe how the test accuracies are near random guessing. This is a verification that without D-BAT, due to the simplicity bias, only the simplest feature is leveraged to predict the label and the models fail to generalize to domains for which the simple feature is spurious. D-BAT however, is effectively promoting models to use diverse features. This is demonstrated by the test accuracies of the best D-BAT model being much higher than of ERM.

- **D-BAT improves generalization to new domains.** In Tab. 1, in the case $\mathcal{D}_{\text{ood}} \neq$ test data, we observe that despite differences between $\mathcal{D}_{\text{ood}}$ and the test distribution (e.g. the target distribution for M/C-D is using CIFAR-10 images of cars and trucks whereas $\mathcal{D}_{\text{ood}}$ uses images of frogs, cats, etc. but no cars or trucks), D-BAT is still able to increase the generalization to the test domain.

- **Improved generalization on natural datasets.** We observe a significant improvement in test accuracy for all our natural datasets. While the improvement is limited for the Office home dataset when considering a single model, we observe D-BAT ensembles nonetheless outperform ERM ensembles. The improvement is especially evident on the Camelyon17 dataset where D-BAT outperforms many known methods as seen in Fig. 4.c.

- **Ensembles built using D-BAT generalize better.** In Fig. 4 we observe how D-BAT ensembles trained on the Waterbirds and Office-Home datasets generalize better.

### 4.2 Better Uncertainty & OOD Detection

**MNIST setup.** We run two experiments to investigate D-BAT's ability to provide good uncertainty estimates. The first one is similar to the MNIST experiment in Liu et al. (2021b), it consists in learning to differentiate MNIST digits 0s from 1s. The uncertainty of the model — computed as the entropy — is then estimated for fake interpolated images of the form $t \cdot 1 + (1 - t) \cdot 0$ for $t \in [-1, 2]$. An ideal model would assign (i) low uncertainty values for $t$ near 0 and 1, corresponding to in-distribution samples, while (ii) high uncertainty values elsewhere. (Liu et al., 2021b) showed how only Gaussian Processes are able to fulfill those two conditions, most models failing in attributing high uncertainty *away* from the boundary decision (as it can also be seen in Fig. 1 when looking at individual models). We train ensembles of size 2 and average over 20 seeds. For D-BAT, we use as $\mathcal{D}_{\text{ood}}$ the remaning (OOD) digits 2 to 9, along with some random cropping. We use a LeNet.

**MNIST results.** Results in Fig. 5 suggest that D-BAT is able to give reliable uncertainty estimates for OOD datapoints, even when those samples are away from the boundary decision. This is in sharp contrast with deep-ensemble which only models uncertainty near the boundary decision.

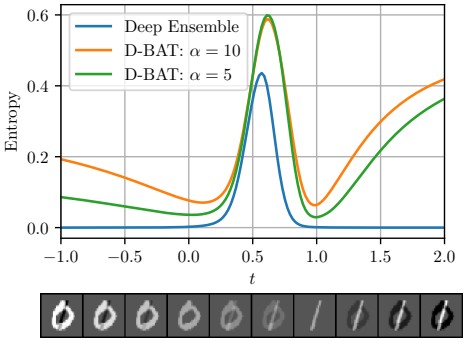

Figure 5: Entropy of ensembles of two models trained with and without D-BAT (deep-ensemble), for inputs $x$ taken from along line $t \cdot 1 + (1-t) \cdot 0$ for $t \in [-1, 2]$. In-distribution samples are obtained for $t \in \{0, 1\}$. All ensembles have a similar test accuracy of $99\%$. Unlike deep ensembles, D-BAT ensembles are able to correctly give high uncertainty values for points far away from the decision boundary. The standard deviations have been omitted here for clarity, but can be seen in App. D.3.

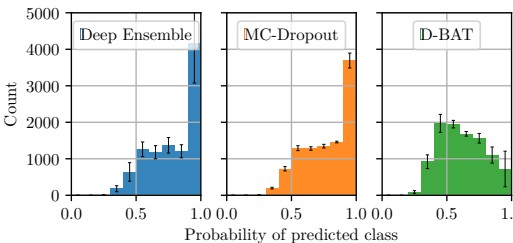

Figure 6: Histogram of predicted probabilities on OOD data. See § 4.2 for more details on the setup. D-BAT ensembles are better calibrated with less confidence on OOD data than deep-ensembles or MC-Dropout models.

**CIFAR-10 setup.** We train ensembles of 4 models and benchmark three different methods in their ability to identify what they do not know. For this we look at the histograms of the probability of their predicted classes on OOD samples. As training set we use the CIFAR-10 classes $\{0, 1, 2, 3, 4\}$. We use the CIFAR-100 (Krizhevsky, 2009) test set as OOD samples to compute the histograms. For D-BAT we use the remaining CIFAR-10 classes, $\{5, 6, 7, 8, 9\}$, as $\mathcal{D}_{ood}$, and set $\alpha$ to $0.2$. Histograms are averaged over 5 seeds. The three methods considered are simple deep-ensembles (Lakshminarayanan et al., 2017), MC-Dropout models (Gal & Ghahramani, 2016), and D-BAT ensembles. For the three methods we use a modified ResNet-18 (He et al., 2016) with added dropout to accommodate MC-Dropout, we use a dropout probability of $0.2$ for the three methods. For MC-Dropout, we compute uncertainty estimates sampling 20 distributions.

**CIFAR-10 results.** In Fig. 6, we observe for both deep ensembles and MC-Dropout a large amount of predicted probabilities larger than $0.9$, which indicate those methods are overly confident on OOD data. In contrast, most of the predicted probabilities of D-BAT ensembles are smaller than $0.7$. The average ensemble accuracies for all those methods are $92\%$ for deep ensembles, $91.2\%$ for D-BAT ensembles, and $90.4\%$ for MC-Dropout.

## 5 LIMITATIONS

Is the simplicity bias gone? While we showed in § 4.1 that our approach can clearly mitigate shortcut learning, a bad choice of $\mathcal{D}_{ood}$ distribution can introduce an additional shortcut. In essence, our approach fails to promote diverse representations when differentiating $\mathcal{D}$ from $\mathcal{D}_{ood}$ is easier than learning to utilize diverse features. Furthermore, we want to stress that learning complex features is not necessarily unilaterally better than learning simple features, and is not our goal. Complex features are better only so far as they can better explain both the train distribution and OOD data. With our approach, we aim to get a diverse yet simple set of hypotheses. Intuitively, D-BAT tries to find the best hypothesis which may be somewhere within the top-k simplest hypotheses, and not necessarily the simplest one which the simplicity bias is pushing us towards.

## 6 CONCLUSION

Training deep neural networks often results in the models learning to rely on shortcuts present in the training data but absent from the test data. In this work we introduced D-BAT, a novel training method to promote diversity in ensembles of predictors. By encouraging disagreement on OOD data, while agreeing on the training data, we effectively (i) give strong incentives to our predictors to rely on diverse features, (ii) which enhance the transferability of the ensemble and (iii) improve uncertainty estimation and OOD detection. Future directions include improving the selection of samples of the OOD distribution and develop stronger theory. D-BAT could also find applications beyond OOD generalization–e.g. (Ţifrea et al., 2021) recently used disagreement for anomaly/novelty detection or to test for biases in our trained models (Stanczak & Augenstein, 2021).

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

## A    SOURCE CODE

Link to the source code to reproduce our experiments: https://github.com/mpagli/Agree-to-Disagree

## B    ALGORITHMS

The D-BAT training algorithm can be applied to both binary and multi-class classification problems. For our experiments on binary classification — as for the Camelyon17, Waterbirds, M/F-D, M/C-D (see § 4.1), and for our MNIST experiments in Fig. 5 — we used Alg. 1. This algorithm assumes a first model $h_1$ has already been trained with e.g. empirical risk minimization, and trains a second model following the algorithm described in § 3.2. For our multi-class experiments — as for the C-MNIST, Office-Home (see § 4.1, and CIFAR-10 uncertainty experiments (see § 4.2), we used Alg. 2. This algorithm is training a full ensemble of size $M$ using D-BAT as described in § 3.2.

---

**Algorithm 1** D-BAT for binary classification

---

**Input:** train data $\mathcal{D}$, OOD data $\mathcal{D}_{\text{ood}}$, stopping time $T$, D-BAT coefficient $\alpha$, learning rate $\eta$, pre-trained model $h_1$, randomly initialized model $h_2$ with weights $\boldsymbol{\omega}_0$, and its loss $\mathcal{L}$.
**for** $t \in 0, \dots, T-1$ **do**
   **Sample** $(\boldsymbol{x}, y) \sim \mathcal{D}$
   **Sample** $\tilde{x} \sim \mathcal{D}_{\text{ood}}$
   $\boldsymbol{\omega}_{t+1} = \boldsymbol{\omega}_t - \eta \nabla_{\boldsymbol{\omega}} \big( \mathcal{L}(h_2, \boldsymbol{x}, y) + \alpha \mathcal{A}(h_1, h_2, \tilde{x}) \big)$
**end for**

---

**Algorithm 2** D-BAT for multi-class classification

---

**Input:** ensemble size $M$, train data $\mathcal{D}$, OOD data $\mathcal{D}_{\text{ood}}$, stopping time $T$, D-BAT coefficient $\alpha$, learning rate $\eta$, randomly initialized models $(h_0, \dots, h_{M-1})$ with resp. weights $(\boldsymbol{\omega}_0^{(0)}, \dots, \boldsymbol{\omega}_0^{(M-1)})$, and a classification loss $\mathcal{L}$.
**for** $m \in 0, \dots, M-1$ **do**
   **for** $t \in 0, \dots, T-1$ **do**
      **Sample** $(\boldsymbol{x}, y) \sim \mathcal{D}$
      **Sample** $\tilde{x} \sim \mathcal{D}_{\text{ood}}$
      $\mathcal{A} \leftarrow 0$
      $\tilde{y} \leftarrow \arg\max h_m(\tilde{\boldsymbol{x}})$
      **for** $i \in 0, \dots, m-1$ **do**
         $\mathcal{A} = \mathcal{A} - \frac{1}{m-1} \log \left( p_{h_i, \tilde{\boldsymbol{x}}}^{(\tilde{y})} \cdot p_{h_m, \tilde{\boldsymbol{x}}}^{(\neg\tilde{y})} + p_{h_i, \tilde{\boldsymbol{x}}}^{(\neg\tilde{y})} \cdot p_{h_m, \tilde{\boldsymbol{x}}}^{(\tilde{y})} \right)$
      **end for**
      $\boldsymbol{\omega}_{t+1}^{(m)} = \boldsymbol{\omega}_t^{(m)} - \eta \nabla_{\boldsymbol{\omega}^{(m)}} \big( \mathcal{L}(h_m, \boldsymbol{x}, y) + \alpha \mathcal{A} \big)$
   **end for**
**end for**

---

**Sequential vs. simultaneous training.**    Nothing prevents the use of the D-BAT objective while training all the predictors of the ensemble simultaneously. While we had some successes in doing so, we advocate against it as this can discard the ERM solution. We found that the training dynamics of simultaneous training have a tendency to generate more complex solutions than sequential training. In our experiments on the 2D toy setting, sequential training gives two models which are both simple and diverse (see Fig. 1), whereas simultaneous training generates two relatively simple predictors but of higher complexity (see Fig. 7), especially it would deprive us from the simplest solution (Fig.1.b). In general as we do not know the spuriousness of the features, the simplest predictor is still of importance.

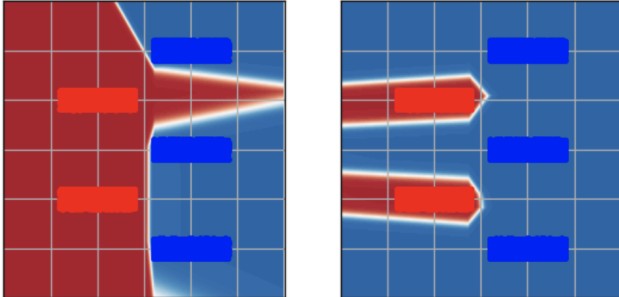

Figure 7: Simultaneous D-BAT training: two models trained simultaneously using D-BAT on our 2D toy task (see Fig. 1). We observe how we do not recover the ERM solution. The two obtained models are diverse but seemingly more complex (e.g. in terms of their boundary decision) than models trained sequentially as in Fig. 1.

## C  PROOF OF THM. 3.1

We redefine here the setup for clarity:

- Given a joint source distribution $\mathcal{D}$ of triplets of random variables $(C, S, Y)$ taking values in $\{0, 1\}^3$.
- Assuming $\mathcal{D}$ has the following pmf: $\mathbb{P}_{\mathcal{D}}(C = c, S = s, Y = y) = 1/2$ if $c = s = y$, and $0$ otherwise.
- Assuming a first model learnt the posterior distribution $\hat{\mathbb{P}}_1(Y = 1 \mid C = c, S = s) = c$.
- Given a distribution $\mathcal{D}_{\text{ood}}$ uniform over $\{0, 1\}^3$ outside of the support of $\mathcal{D}$.

From there, training a second model $h_2$ following the D-BAT objective would mean minimizing the agreement on $\mathcal{D}_{\text{ood}}$:

$$\min_{(c,s)\sim\mathcal{D}_{\text{ood}}} \mathbb{E} \left[ -\log(\hat{\mathbb{P}}_1(Y = 1|c, s)\hat{\mathbb{P}}_2(Y = 0|c, s) + \hat{\mathbb{P}}_1(Y = 0|c, s)\hat{\mathbb{P}}_2(Y = 1|c, s)) \right] \quad (1)$$

While at the same time agreeing on the source distribution $\mathcal{D}$:

$$\mathbb{P}_{(c,s)\sim\mathcal{D}} \left( \hat{\mathbb{P}}_1(Y|c, s) = \hat{\mathbb{P}}_2(Y|c, s) \right) = 1$$

The expectation in eq. 1 becomes:

$$(1) = \frac{1}{2} \left( -\log(\hat{\mathbb{P}}_2(Y = 0|C = 1, S = 0)) - \log(\hat{\mathbb{P}}_2(Y = 1|C = 0, S = 1)) \right)$$

Which is minimized for $\hat{\mathbb{P}}_2(Y = 1|C = 0, S = 1) = \hat{\mathbb{P}}_2(Y = 0|C = 1, S = 0) = 1$.

Which means the posterior of the second model, according to our disagreement constrain, will be:

$$\hat{\mathbb{P}}_2(Y = 1 \mid C = c, S = s) = s$$

## D  OMITTED DETAILS ON EXPERIMENTS

### D.1  IMPLEMENTATION DETAILS FOR THE C-MNIST, M/M-D, M/F-D AND M/C-D EXPERIMENTS

In the experiments on C-MNIST, M/F-D and M/C-D, we used different versions of LeNet (Lecun et al., 1998):

- For the C-MNIST dataset, we used a standard LeNet, with 3 input channels instead of 1.

- For the MF-Dominoes datasets, we increase the input dimension of the first fully-connected layer to 960.
- For the MC-Dominoes dataset, we use 3 input channels, increase the number of output channels of the first convolution to 32, and of the second one to 56. We modify the fully-connected layers to be $2016 \rightarrow 512 \rightarrow 256 \rightarrow c$ with $c$ the number of classes.

In those experiments — for both cases $\mathcal{D}_{\text{ood}} = \mathcal{D}_{\text{test}}$ and $\mathcal{D}_{\text{ood}} \neq \mathcal{D}_{\text{test}}$ — the test and validation distributions are distributions in which the spurious feature is random, e.g. random color for C-MNIST and random $0$ or $1$ on the top part for MF-Dominoes and MC-Dominoes.

We use the AdamW optimizer Loshchilov & Hutter (2019) for all our experiments. For all the datasets in this section, we only train ensembles of 2 models, which we denote $\mathcal{M}_1$ and $\mathcal{M}_2$. When building the OOD datasets, we make sure the images used are not shared with the images used to build the training, test and validation sets. Our results are obtained by averaging over 5 seeds. For further details on the implementation, we invite the reader to check the source code, see § A.

### D.2 IMPLEMENTATION DETAILS FOR FIG. 1

Instead of relying on an external OOD distribution set, it is also possible to find, given some datapoint $\boldsymbol{x}$, a perturbation $\delta^\star$ through directly minimizing the agreement in some neighborhood of $\boldsymbol{x}$ (i.e. for $\|\delta^\star\| \leq \epsilon$):

$$\delta^\star \in \underset{\delta \text{ s.t. } \|\delta\| < \epsilon}{\arg\min} - \log \left( p_{h_1,(\boldsymbol{x}+\delta)}^{(0)} \cdot p_{h_2,(\boldsymbol{x}+\delta)}^{(1)} + p_{h_1,(\boldsymbol{x}+\delta)}^{(1)} \cdot p_{h_2,(\boldsymbol{x}+\delta)}^{(0)} \right)$$

Which can be solved using several projected gradient descent steps as it done typically in the adversarial training literature. While this approach is working for the 2D example, it is not working however for complex high-dimensional input spaces combined with deep networks as those are notorious for their sensitivity to very small $l_p$-bounded perturbations, and it would most of the time be easy to find a bounded perturbation maximizing the disagreement.

### D.3 STANDARD DEVIATIONS FOR MNIST UNCERTAINTY EXPERIMENTS

For clarity we omitted the standard deviations in Fig. 5. In Fig. 8 we show each individual curve with its associated standard deviation.

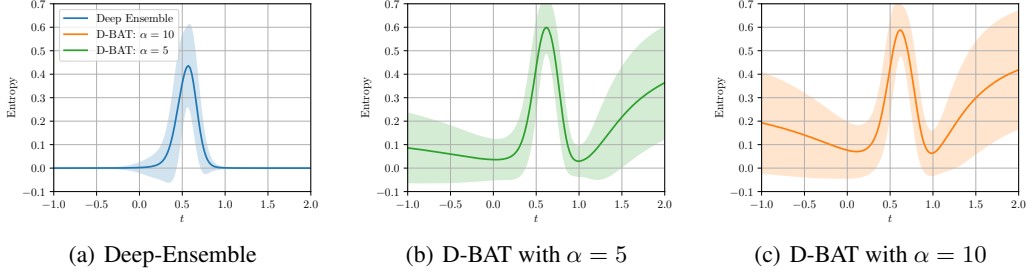

(a) Deep-Ensemble      (b) D-BAT with $\alpha = 5$      (c) D-BAT with $\alpha = 10$

Figure 8: Entropy of ensembles of two models trained with (**(b)** and **(c)**) and without D-BAT (deep-ensemble, **(a)**), for inputs $x$ taken from along line $t \cdot 1 + (1-t) \cdot 0$ for $t \in [-1, 2]$. For deep-ensembles in **(a)**, we notice how the standard deviation is near 0 for OOD regions $t \in ]-1, 0] \cup [1, 2[$, which indicates a lack of diversity between members of the ensemble. This is in sharp contrast with D-BAT ensembles in **(b)** and **(c)** which clearly show some variability in those regions. The high variability is explained by the fact that we are not optimizing specifically to be able to detect OOD samples in those regions, but instead we are gaining this ability as a by-product of diversity, and diversity can be reached in many different configurations.

### D.4 IMPLEMENTATION DETAILS FOR THE CAMELYON17 EXPERIMENTS

The CameLyon17 cancer detection dataset Bandi et al. (2018) is taken from the WILDS collection Koh et al. (2021). The dataset consists of a training, validation, and test sets of images coming from

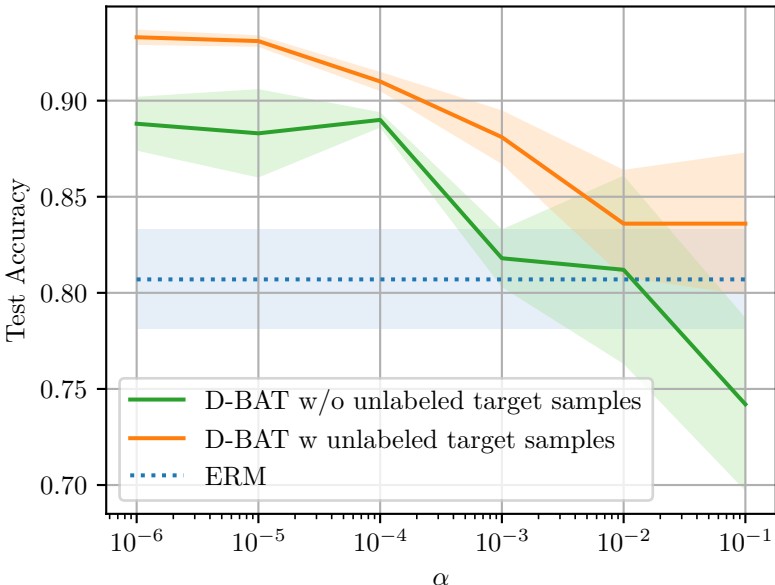

Figure 9: Test accuracy given $\alpha$. We compare the best ERM model with the second model trained using D-BAT, for varying $\alpha$ hyperparameters.

different hospitals, each hospital being uniquely associated to a given split. The goal is to generalize to hospitals not necessarily present in the training set.

In the case where $\mathcal{D}_{\text{ood}} = \mathcal{D}_{\text{test}}$, we use the *unlabeled* test data provided by WILDS as $\mathcal{D}_{\text{ood}}$. In our experiments with $\mathcal{D}_{\text{ood}} \neq \mathcal{D}_{\text{test}}$, we use the *unlabeled* validation data provided by WILDS as $\mathcal{D}_{\text{ood}}$. In both cases we use the accuracy on the WILDS labeled validation set for model selection.

We use a ResNet-50 (He et al., 2016) as model. We train for 60 epochs with a fixed learning rate of 0.001 with and SGD as optimizer. We use an $l_2$ penalty term of 0.0001 and a momentum term $\beta = 0.9$. For D-BAT, we tune $\alpha \in \{10^{-1}, 10^{-2}, 10^{-3}, 10^{-4}, 10^{-5}, 10^{-6}\}$ and found $\alpha = 10^{-6}$ to be best. For each set of hyperparameters, we train a deep-ensemble and a D-BAT ensemble of size 2, and select the parameters associated with the highest averaged validation accuracy over the two predictors of the ensemble. Our results are obtained by averaging over 3 seeds.

In Fig. 9, we plot the evolution of the test accuracy as a function of $\alpha$ for both setups discussed in § 4.1. In the first "ideal" setup we have access to unlabeled target data to use as $\hat{\mathcal{D}}_{\text{ood}}$. In the second setup we do not, instead we use samples from different hospitals. In the case of the Camelyon dataset, we use the available unlabeled validation data. Despite this data belonging to a different domain, we still get a significant improvement in test accuracy.

## D.5 Implementation details for the Waterbirds experiments

The Waterbirds dataset is built by combining images of birds with either a water or land background. It contains four categories:

- Waterbirds on water
- Waterbirds on land
- Land-birds on water
- Land-birds on land

In the official version released in the WILDS suite, the background is predictive of the label in 95% of cases i.e. 95% of Waterbirds, resp. land-birds, are seen on water, resp. land. Due to the simplicity bias, this means that ERM models tend to overuse the background information. The test

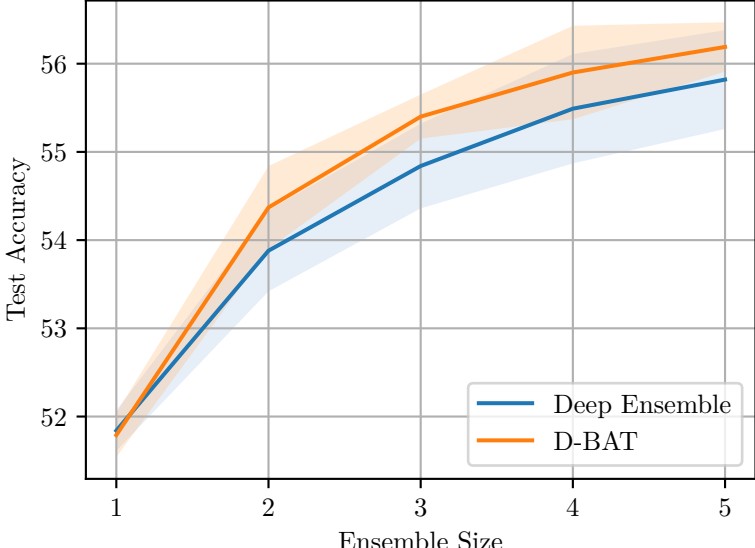

Figure 10: Comparing test accuracy for D-BAT and ERM (Deep-ensemble) for different ensemble sizes. For D-BAT, $\mathcal{D}_{\text{ood}}$ is the "Art" domain, quite different from the "Real-world" domain. Despite the distribution shift we still see a noticeable improvement using D-BAT over plain ERM.

and validation sets are made more evenly, with $50\%$ of Waterbirds, resp. land-birds, being seen on water, resp. land. We use the train/ validation/test splits provided by the WILDS library.

We use a ResNet-50 (He et al., 2016) as model. We train for 300 epochs with a fixed learning rate of $0.001$ with and SGD as optimizer. We an $l_2$ penalty term of $0.0001$ and a momentum term $\beta = 0.9$. For D-BAT, we tune $\alpha \in \{10^0, 10^{-1}, 10^{-2}, 10^{-3}, 10^{-4}, 10^{-5}\}$ and found $\alpha = 10^{-4}$ to be best. For each set of hyperparameters, we train a deep-ensemble and a D-BAT ensemble of size 2, and select the parameters associated with the highest averaged validation accuracy over the two predictors of the ensemble. Our results are obtained by averaging over 3 seeds.

For our D-BAT experiments we only consider the case where we have access to unlabeled target data. We use the validation split as it is from the same distribution as the target data.

### D.6 IMPLEMENTATION DETAILS FOR THE OFFICE-HOME EXPERIMENTS

The Office-Home dataset is made of four domains: Art, Clipart, Product, and Real-world. We train on the grouped Product and Clipart domains, and measure the generalization to the Real-world domain. This dataset has 65 classes.

We use a ResNet-18, we train for 600 epochs with a fixed learning rate of $0.001$ with and SGD as optimizer. We an $l_2$ penalty term of $0.0001$ and a momentum term $\beta = 0.9$. For D-BAT, we tune $\alpha \in \{10^0, 10^{-1}, 10^{-2}, 10^{-3}, 10^{-4}, 10^{-5}, 10^{-6}\}$ and found $\alpha = 10^{-5}$ to be best. For each set of hyperparameters, we train a deep-ensemble and a D-BAT ensemble of size 2, and select the parameters associated with the highest averaged validation accuracy over the two predictors of the ensemble. Our results are obtained by averaging over 6 seeds.

We experiment with both the "ideal" case in which some unlabeled target data is available to use as $\mathcal{D}_{\text{ood}}$ ($\mathcal{D}_{\text{ood}} = \mathcal{D}_{\text{test}}$; see Fig. 4.b) as well as the case in which we use a different domain (Art) as $\mathcal{D}_{\text{ood}}$ ($\mathcal{D}_{\text{ood}} \neq \mathcal{D}_{\text{test}}$). For this later setup, the evolution of the test accuracy given the ensemble size is in Fig. 10. In both cases, the validation split, just as the test split, comes from the Real-World domain.

### D.7 NOTE ON SELECTING $\alpha$

Depending on the experiment the value of $\alpha$ used ranged from 1 to $10^{-6}$. We explain the variability in those values by (i) the capacity of the model used and (ii) the OOD distribution selected. If the model used has a large capacity, it can more easily overfit the OOD distribution and find shortcuts to disagree on $\mathcal{D}_{\text{ood}}$ without relying on different features to classify the training samples, as discussed in § 5. For this reason we observed that larger models such as ResNet-18 or ResNet-50 used respectively on CIFAR10 and the Camelyon17 datasets are requiring a smaller $\alpha$ in comparison to smaller LeNet architectures. Furthermore, when the OOD distribution is close to the training distribution, smaller $\alpha$ values are preferred, as in our Camelyon17 experiments. In this case, disagreeing too strongly on the OOD data might force a second model $\mathcal{M}_2$ to give erroneous predictions to disagree with $\mathcal{M}_1$, assuming that this first model is generalizing well to the OOD set.

### D.8 COMPUTATIONAL RESOURCES

All of our experiments were run on single GPU machines. Most of our experiments require little computational resources and can be entirely reproduced on e.g. google colab (see App. A). For the Camelyon17, Waterbirds and Office-Home datasets, which use a ResNet-50 or ResNet-18 architectures, we used a V100 Nvidia GPU and the hyperparameter search and training took about two weeks.

## E TRAINING AND OOD DISTRIBUTION SAMPLES C-MNIST, M/F-D AND M/C-D

In Fig. 11, we show some samples from some of the training distribution used in § 4.1. We also introduce the MM-Dominoes dataset, similar in spirit to the other dominoes dataset but concatenating MNIST digits of 0s and 1s with MNSIT digits 7 and 9. In Figs. 12,13,14, we show samples for the OOD distributions used in § 4.1.

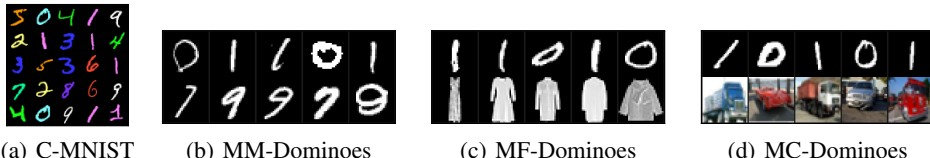

(a) C-MNIST     (b) MM-Dominoes     (c) MF-Dominoes     (d) MC-Dominoes

Figure 11: Samples from the training data distribution $\mathcal{D}$ for C-MNIST, MM-Dominoes, MF-Dominoes, and MC-Dominoes. Those datasets are used to evaluate D-BAT's aptitude to evade the simplicity bias. For C-MNIST, the simple feature is the color and the complex one is the shape. For all the Dominoes datasets, the simple feature is the top row, while the complex feature is the bottom one. One could indeed separate 0s from 1s by simply looking at the value of the middle pixels (if low value then 0 else 1).

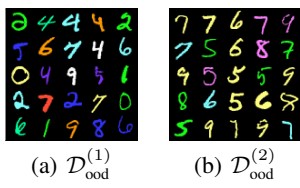

(a) $\mathcal{D}_{\text{ood}}^{(1)}$     (b) $\mathcal{D}_{\text{ood}}^{(2)}$

Figure 12: OOD distributions used for the C-MNIST experiments. $\mathcal{D}_{\text{ood}}^{(1)}$ is the distribution used to train D-BAT when we assumed we have access to unlabeled target data. $\mathcal{D}_{\text{ood}}^{(2)}$ is the distribution we used to show how D-BAT could work despite not having unlabeled target data. When experimenting on $\mathcal{D}_{\text{ood}}^{(2)}$ we remove the shapes 5 to 9 from the training dataset, that way $\mathcal{D}_{\text{ood}}^{(2)}$ is really OOD.

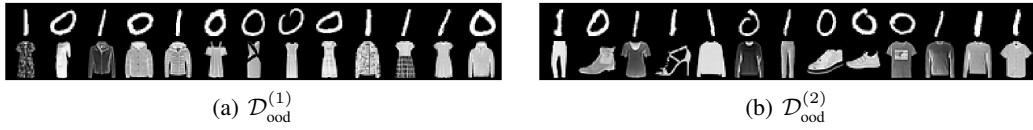

(a) $\mathcal{D}_{\text{ood}}^{(1)}$ (b) $\mathcal{D}_{\text{ood}}^{(2)}$

Figure 13: OOD distributions used for the MF-Dominoes experiments. $\mathcal{D}_{\text{ood}}^{(1)}$ corresponds to our experiments when we have access to unlabeled target data. $\mathcal{D}_{\text{ood}}^{(2)}$ is very different from the target distribution as the second row is made only of images from categories not present in the training and test distributions.

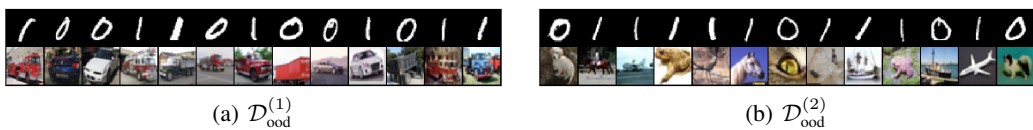

(a) $\mathcal{D}_{\text{ood}}^{(1)}$ (b) $\mathcal{D}_{\text{ood}}^{(2)}$

Figure 14: OOD distributions used for the MC-Dominoes experiments. $\mathcal{D}_{\text{ood}}^{(1)}$ corresponds to our experiments when we have access to unlabeled target data. $\mathcal{D}_{\text{ood}}^{(2)}$ is very different from the target distribution as the second row is made only of images from categories not present in the training and test distributions.

## F ADDITIONAL DISCUSSIONS AND EXPERIMENTS

When two features are equally predictive but have different complexities, the more complex feature will be discarded due to the extreme simplicity bias. This happens despite the uncertainty over the potential spuriousness of the simpler feature. For this reason it is important to be able to learn both features if we hope to improve our chances at OOD generalization. Recent methods such as Saito et al. (2017), Saito et al. (2018), Zhang et al. (2021), Nam et al. (2020) and Liu et al. (2021a) all fail in this challenging scenario, we explain why in the following subsections F.1 to F.6. In F.7, we add a comparison between D-BAT and the concurrent work of Lee et al. (2022).

### F.1 COMPARISON WITH TENEY ET AL. (2021)

In their work, Teney et al. (2021) add a regularisation term $\boldsymbol{\delta}_{g_{\boldsymbol{\varphi}_1}, g_{\boldsymbol{\varphi}_2}}$ which, given an input $\boldsymbol{x}$, is promoting orthogonality of hidden representations $\boldsymbol{h} = f_{\boldsymbol{\theta}}(x)$ given by an encoder $f_{\boldsymbol{\theta}}$ with parameters $\boldsymbol{\theta}$, and pairs of classifiers $g_{\boldsymbol{\varphi}_1}$ and $g_{\boldsymbol{\varphi}_2}$ of parameters $\boldsymbol{\varphi}_1$ and $\boldsymbol{\varphi}_2$ respectively:

$$\boldsymbol{\delta}_{g_{\boldsymbol{\varphi}_1}, g_{\boldsymbol{\varphi}_2}} = \nabla_{\boldsymbol{h}} g_{\boldsymbol{\varphi}_1}^{\star}(x) \cdot \nabla_{\boldsymbol{h}} g_{\boldsymbol{\varphi}_2}^{\star}(x) \tag{T}$$

With $\nabla g^{\star}$ the gradient of its top predicted score.

We implemented the objective of Teney et al. (2021) with two different encoders: $f_{\boldsymbol{\theta}}(x) = x$ (identity) and a two-layers CNN. We tested it on our MM-Dominoes dataset (See App E). The classification heads are trained simultaneously. Considering two classifications heads, we find two sets of hyperparameters, one that is giving the best compromise between accuracy and randomized-accuracy, and one that is keeping the accuracy close to 1. In the first setup in Fig. 15, we observe that none of the pairs of models trained with equation T as regularizer are particularly good at capturing any of the two features in the data. In contrast with D-BAT (with $\mathcal{D}_{\text{ood}}^{(1)}$) which is able to learn a second model having both high accuracy and high randomized-accuracy, hence capturing with the first model the two data modalities. For the second set of hyperparameters in Fig. 16, we observe that the improvement in randomized accuracy is only marginal if we do not want to sacrifice accuracy. We believe those results are explained by the many ways gradients of a neural network can be orthogonal while still encoding identical information. Better results might require training more classification heads (up to 96 heads are used in Teney et al. (2021).

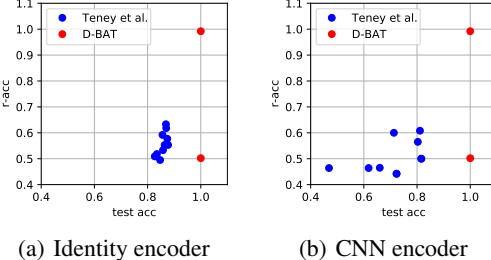

(a) Identity encoder      (b) CNN encoder

Figure 15: Comparison between D-BAT and Teney et al. (2021) with hyperparameters favoring the compromise between accuracy (test-acc) and randomized-accuracy (r-acc). We run 5 different seeds for Teney et al. (2021), each run consisting in two classification heads and a shared encoder chosen to be the identity (a) or a CNN encoder (b). The acc and r-acc are displayed for the 10 resulting classification heads. We compared with two models obtained using D-BAT, the first model learning the simplest feature is in the bottom right corner, and the second model trained with diversity is in the top right corner. We observe that the method of Teney et al. (2021) is failing to reach a good r-acc, and is sacrificing accuracy. D-BAT is able to retrieve both data modalities without sacrificing accuracy.

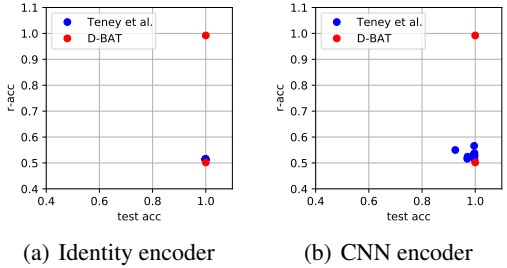

(a) Identity encoder      (b) CNN encoder

Figure 16: Comparison between D-BAT and Teney et al. (2021) with hyperparameters yielding an accuracy (test-acc) close to 1 while maximizing the randomized-accuracy (r-acc). We run 5 different seeds for Teney et al. (2021), each run consisting in two classification heads and a shared encoder chosen to be the identity (a) or a CNN encoder (b). The acc and r-acc are displayed for the 10 resulting classification heads. We compared with two models obtained using D-BAT, the first model learning the simplest feature is in the bottom right corner, and the second model trained with diversity is in the top right corner. We observe that the method of Teney et al. (2021) is only marginally improving the randomized-acc.

### F.2 COMPARISON WITH ZHANG ET AL. (2021)

In their work, Zhang et al. (2021) argue that while a model can be biased, there exist unbiased functional subnetworks. They introduced Modular Risk Minimization (MRM) to find those subnetworks. We implemented the MRM method (Alg.1 from their paper) and tested it on our MM-Dominoes dataset (§ 4.1). We observed that their approach cannot handle the extreme case we consider where the spurious feature is fully predictive in the train distribution (but not in OOD). They need it to be, say, only 90% predictive. On our dataset, in the first phase of Alg.1, the model trained on the source task learns to completely ignore the bottom row due to the extreme simplicity bias, ensuring there is no useful sub-network. We found the randomized-accuracy of subnetworks obtained with MRM to be no better than random. This is because, in extreme cases, the network which the simplicity bias pushes us to learn may completely ignore the actual feature and instead only focuses on the spurious feature. In such a case, there is no un-biased subnetwork.

### F.3 COMPARISON WITH SAITO ET AL. (2017)

Contrary to Saito et al. (2017), we aim to train an ensemble of predictors able to generalize to *unknown* target tasks and do not assume access to the target data. In particular, the unlabelled OOD data we need can be different from the downstream transfer target data. We make this distinction clear in § 4.1 where $\mathcal{D}_{\text{ood}}^{(3)}$ for the dominoes datasets are built using combinations of 1s and 0s with images from classes *not present* in the target and source tasks. Despite the lack of target data, the r-acc improves by resp. 28% and 38% for the MM-Dominoes and MF-Dominoes datasets. Further, we focus on mitigating extreme *simplicity bias* as described by Shah et al. (2020), where a spurious feature can have the same predictive power as a non-spurious one on the source task (but not on the unknown target task). While (Saito et al., 2017) uses the concept of diversity, their formulation measures diversity in temrs of the inner-product between the weights. However, since neural networks are highly non-convex, it is possible for two networks to effectively learn the exact same function which relies on spurious features, while still having different parameterization. Thus, our method can be viewed as "functional" extension of the method in (Shah et al., 2020). Further, the encoder $F$ itself can learn a representation such that $F_1$ and $F_2$ rely on the same information while minimizing the regularizer.

To see this, we trained the method of Alg.1 from (Saito et al., 2017) on our MM-Dominoes dataset. Tuning $\lambda \in \{0.1, 1, 10, 100\}$, we were unable to learn a model $F_t$ which transfers to the target task.

### F.4 COMPARISON WITH SAITO ET AL. (2018)

Contrary to Saito et al. (2018), we do not aim at training a domain agnostic representation, but instead on overcoming simplicity bias to generalize to OOD settings. E.g. in colored MNIST, a classifier which throws out the shape and simply uses color (or vice-versa) is domain agnostic. But for overcoming spurious features, models in our ensemble would need to use *both* color and digit. Thus a domain agnostic representation is insufficient for OOD generalization.

Furthermore, the training procedure of (Saito et al., 2018) consists in first training a shared feature extractor $G$ and two classification heads $F_1$ and $F_2$ to minimize the cross-entropy on the source task. In a second step the classification heads $F_1$ and $F_2$ are trained to increase the discrepancy on samples from the target distribution while fixing the feature extractor $G$. However, in the case where a spurious feature is as predictive as the non-spurious one — as in our experiments of § 4.1 — the extreme simplicity bias would force the feature extractor to become invariant to the complex feature. The second and third steps of the algorithm would fail from there.

### F.5 COMPARISON WITH NAM ET AL. (2020)

In this work, two models are trained simultaneously, one being the biased model while the other is the debiased model. During training, the first model gives higher weights to training samples agreeing with the current bias of the model. On the other hand, the second model learns by giving higher weights to training samples conflicting with the biased model. In order to work, the algorithm considers that the ratio of bias-aligned samples is smaller than 100%, which is not the case for our datasets in § 4.1). In these challenging datasets, where the biased feature is as predictive as the not

biased feature, the second model fails to find bias-conflicting samples, hence would fail to de-biased itself. For this reason, the work of Nam et al. (2020) fails to counter extreme simplicity bias.

## F.6 COMPARISON WITH LIU ET AL. (2021A)

The work of Liu et al. (2021a) is similar to the work of Nam et al. (2020) and shares the same limitation. A first model is trained through ERM before a second model trained by upweighting the samples misclassified in by the first model. This method, as for Nam et al. (2020), is failing to induce diversity when all the samples are correctly classify by the first model, as this is the case for our datasets in § 4.1.

We implemented the JTT method from Liu et al. (2021a) and report test accuracies on the Waterbird, Camelyon17, and Office-Home datasets in Table 2. We tuned $T$, the number of epochs for the first model, in $\{1, 2, 5, 10, 20, 60\}$. We tune the upsampling weight $\lambda$ in $\{6, 50, 100\}$. We pick the model with best validation accuracy.

Table 2: Comparison between ERM, D-BAT, and JTT. For JTT, results are reported for a single seed. While JTT is efficient when small sub-groups are present in the data —as it is the case in the Waterbirds dataset— the method fails to significantly improve upon ERM when the distribution shift is more severe as in the Office-Home and Camelyon17 datasets.

| Method | Waterbirds | Office-Home | Camelyon17 |
|---|---|---|---|
| ERM | 86.0 | 50.4 | 80.3 |
| JTT | 91.6 | 49.3 | 81.0 |
| D-BAT (ours) | 88.7 | 51.1 | 93.1 |

## F.7 COMPARISON WITH LEE ET AL. (2022)

The concurrent work of Lee et al. (2022) proposes to measure diversity between two models using the mutual information (MI) between their predictions on the entire OOD distribution, whereas our loss is defined on the per datapoint difference in the predictions. This means that our loss decomposes as a sum over the data-points and is well defined on small mini-batches. Computing the mutual information (MI) needs processing the entirety(or at least a very large part) of the data. Besides such practical advantages, our notion of diversity naturally arises out of discrepancy based domain adaptation theory, whereas the choice of using MI is ad-hoc and in fact may not give the expected results. Consider the toy-problem in Fig.3 of Lee et al. (2022) - the predictions of the two models actually have maximum mutual information since they predict the exact opposite on all the unlabelled perturbation data. Thus, MI would say that the two models actually have zero diversity, whereas discrepancy would say they have very high diversity. Hence, MI is theoretically the wrong measure to use. We confirmed this intuition by running experiments on the same setup as in Lee et al. (2022), we compared for the two notions of diversity (MI and discrepancy) which pairs of predictor are optimal. Results can be seen in Fig. 17.

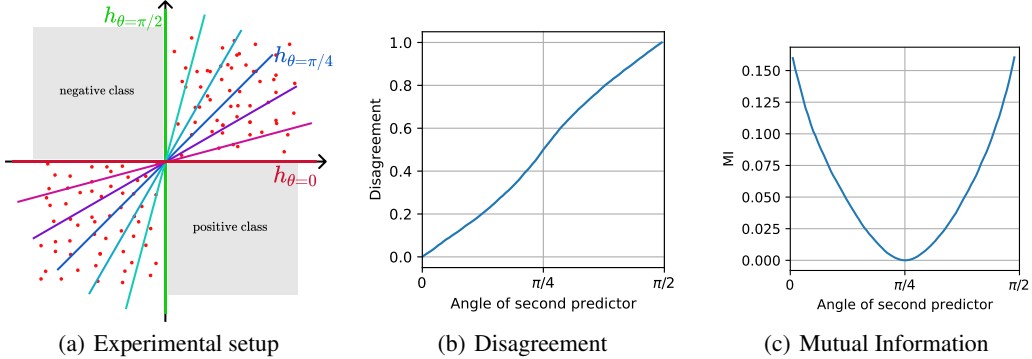

(a) Experimental setup        (b) Disagreement        (c) Mutual Information

Figure 17: Disagreement and mutual information of potential second models $h_2$. In **(a)** we summarize the experimental setup which is similar to Fig.3 of Lee et al. (2022). The training data consists of the diagonal regions of $[0, 1] \times [-1, 0]$ as class 1 (positive), and $[-1, 0] \times [0, 1]$ as class 2 (negative). OOD datapoints $\tilde{X}$ are sampled randomly in the off-diagonal $[-1, 0]^2$ and $[0, 1]^2$ regions. The set of hyperplanes $h_\theta$ with $\theta \in [0, \pi/2]$ all achieve a perfect train accuracy. We fix the first classifier to be the horizontal $h_1 = h_{\theta=0}$ classifier. Then, we measure the disagreement between $h_1$ and different choices of $h_2 = h_\theta$ (in **b**), as well as their mutual information (in **c**) using the code provided in (Lee et al., 2022). Maximizing the disagreement yields the correct vertical classifier $h_2 = h_{\theta=\frac{\pi}{2}}$, whereas minimizing mutual information would yield the *wrong diagonal classifier*. The disagreement scores match intuitive definitions of diversity, whereas mutual information does not.

