# OpenReview forum: "Agree to Disagree: Diversity through Disagreement for Better Transferability"
_ICLR.cc/2023/Conference — ICLR 2023 notable top 5%_

### Official Review · Reviewer_3kp3 · 2022-10-21

**Confidence:** 4
**Correctness:** 3
**Technical Novelty And Significance:** 3
**Empirical Novelty And Significance:** 4
**Recommendation:** 8

**Clarity, Quality, Novelty And Reproducibility:**

The paper is clear and there is obviously a lot of work behind it.
The contributions are fairly novel and the addressed problem is very relevant to the community.
A link to the code repo is provided, thus the method is easy to reproduce.

**Strength And Weaknesses:**

Pros :
- The paper is very well written and clear.
- The claims are supported by both theoretical arguments and empirical evidence
- The methods successfully avoid spurious correlations

Cons :
- Some parts of the paper are a too quick on some choices (agreement loss term, number of learners in the ensemble, weights of the ensemble ..)


**Summary Of The Paper:**

This paper adresses OOD generalization by learning an ensemble of diverse predictors. Diversity can be very handy to overcome distribution shifts between a training and a test set. Indeed, if a model has learnt a spurious correlation, a second will be likely not to, due to the diversity constraint. Diversity is enforced by adding an « agreement » term between individual models in the loss function that is minimized during training. The sought disagreement is enforced on OOD (unannotated) inputs. At inference time, the average of the trained models is used to issue predictions.

**Summary Of The Review:**

My most important remark is concerned with the definition of the ensemble model $h_{ens}$. Information in this respect in the paper is really insufficient. On page 2,  the authors say « the ensemble can either be used by forming a weighted average of the probability distribution from each hypothesis, or by tuning the weights on a downstream task. » but later in the text they give no information on actual mechanisms to decide on these weights.
It is possible that a majority of trained learners will not perform very well on OOD data so a mere average aggregation seems risky to me. In the experimental section, the authors say «Among the two models of the ensemble, the best model is selected according to its validation accuracy. ». While I agree that section is a special form of weighted averaging, the philosophy is a bit different and should be stated earlier in the paper. Also, what is this validation data ? Annotated OOD ones ?

The choice of the agreement loss term is not sufficiently justified in the main body of the article. More details can be found in the appendices (F.7) but I was expecting more context on this choice.

How does the approach scale to increasingly many learners in the ensemble ? Most of the text and experiments use only two learners.  Fig 10 shows an experiment with more learners but again it is in the appendices. Is there a stopping rule to know that sufficiently many learners have been trained.

In the experiment of Fig. 1, OOD points are generated through adversarial perturbations. In the case where OOD points are not available, is this way to generate OOD data recommended ?

Have the authors considered a scenario where the ensemble is trained on a pair of distributions (training and OOD one in the paper notations) but is evaluated on a third one ? In other words, does D-BAT also improve transferability ?

Minor : some typos

Page 3 : « set constrains the output » -> « set constrains on the output »
Page 5 : « to minimize our training data » -> « to minimize risk on our training data »

---

> ### Author Response · Authors · 2022-11-14
> **Response to reviewer 3kp3**
>
> Thank you for taking the time to review our paper and for your feedback!
>
> > While I agree that section is a special form of weighted averaging, the philosophy is a bit different and should be stated earlier in the paper.
>
> In our paper we each time, either select a single model or use the ensemble with equal weights. Whether to go for one or the other solution depends on the data available. As we usually have labeled validation data, we can use this to decide whether to pick a single model or the ensemble. We report the ensemble for most of our experiments to show that even if we do not have labeled validation data, selecting the ensemble is often a conservative choice. This being said it is true that we did not use an optimization algorithm to tune the weights of models in the ensemble. We now clarified this in the introduction.
>
> > what is this validation data ? Annotated OOD ones ?
>
> We now clarified which validation data is being used for all the datasets in the appendix (App.D).
>
> > The choice of the agreement loss term is not sufficiently justified in the main body of the article. More details can be found in the appendices (F.7) but I was expecting more context on this choice.
>
> Interestingly, in the binary classification case, we can derive the disagreement term inside the log from the expected loss, assuming the 01-loss, and assuming independence between $h_1$ and $h_2$:
> $$\\mathbb{E}[C(h_1(x)) \neq C(h_2(x))]  =  h_1(x)(1-h_2(x)) + (1-h_1(x)) h_2(x)$$
> Where $C:[0,1] \mapsto \\{-1,+1\\}$ takes the probability and returns the class. We added a similar comment in the main paper.
>
> > How does the approach scale to increasingly many learners in the ensemble ?
>
> Fig.4.a and 4.b also show the accuracy as we increase the number of learners. The number of learners required seems to be varying depending on the dataset. One heuristic could be to pick the size of the ensemble proportional to the number of simple features present in the data.
>
> > In the experiment of Fig. 1, OOD points are generated through adversarial perturbations. In the case where OOD points are not available, is this way to generate OOD data recommended ?
>
> Using adversarial perturbations is an excellent approach when we know how to sample such perturbations from within the image manifold. We found that naively using adversarial perturbations as OOD data wasn’t very useful beyond trivial settings since they always tend to go outside the manifold. Instead, if unlabelled OOD data is absent, we would recommend using smarter augmentations such as blurs, masks, rotations, etc.
>
> > Have the authors considered a scenario where the ensemble is trained on a pair of distributions (training and OOD one in the paper notations) but is evaluated on a third one ?
>
> Yes, results for this scenario are in the "$\mathcal{D}_\text{ood} \\neq \text{test data}$" column in Tab.1. We observe that the improvements are still significant. For instance, on the Camelyon17 dataset, disagreeing on unlabeled validation samples instead of unlabeled test samples (different hospitals) increases the accuracy from $80.3$ to $88.8$.

---

> > ### Comment · Reviewer_3kp3 · 2022-11-20
> > **Comment**
> >
> > Thanks a lot for all the clarifications.

---

### Official Review · Reviewer_Yjzs · 2022-10-24

**Confidence:** 4
**Correctness:** 3
**Technical Novelty And Significance:** 4
**Empirical Novelty And Significance:** 4
**Recommendation:** 8

**Clarity, Quality, Novelty And Reproducibility:**

**Clarification questions/suggestions**
- Abstract should make it clear that this paper only leverages unlabeled OOD data with their DBAT objective.

**Strength And Weaknesses:**

**Strengths**
- This paper tackles an important problem and is nicely written. The paper is easy to follow and very clear in its description in every section
- Nice motivation for the proposed method is provided in Section 3
- Experimental evaluation is interesting. The proposed method is evaluated for diverse tasks ) including mitigating shortcut learning,
bypassing simplicity bias, and generalizing to OOD distributions.
- All the experimental details and code is available to reproduce experiments from the paper
- Interesting discussion highlighting the weaknesses and limitations of the paper is provided in Section 5

**Weakness**
- Discussion on how should one choose the weight $\alpha$. While the paper presents some results, it is unclear how this hyperparameter is picked in the experiments. It would also be good to discuss the impact of this hyperparameter in different settings for practitioners. Is there a general recipe that authors observed while choosing different weight parameters $\alpha$.

**Summary Of The Paper:**

This paper seeks to tackle the simplicity bias issue of GD algorithms highlighted in prior work. To this end, the authors propose the Diversity-By-disAgreement Training (D-BAT) objective to learn predictors that make diverse predictions on OOD unlabeled data while agreeing on the labeled in-distribution data. Experiments across several different benchmark datasets show the efficacy of the proposed objective.

**Summary Of The Review:**

Overall, the paper makes an interesting contribution and is nicely written. The proposed D-BAT objective is well-motivated with an interesting set of experiments across diverse tasks. I recommend acceptance of the paper.

---

> ### Author Response · Authors · 2022-11-14
> **Response to Yjzs**
>
> Thank you for the valuable time spent reviewing our paper!
>
> > Discussion on how should one choose the weight α
>
> We give some intuition on the choice of $\\alpha$ in App.D.7. In short, we identified two main factors: (i) the spuriousness of the features and (ii) the capacity of the neural architecture. The more spurious the feature is, the larger the $\\alpha$. Larger models can more easily overfit the OOD distribution and are usually favoring smaller $\\alpha$ values (App.D.7 gives more details). In our experiments we typically tune alpha through a log scan.

---

> > ### Comment · Reviewer_Yjzs · 2022-11-18
> > **Thanks for your rebuttal**
> >
> > Thank you for your clarification.

---

### Official Review · Reviewer_nyu8 · 2022-10-24

**Confidence:** 3
**Correctness:** 3
**Technical Novelty And Significance:** 3
**Empirical Novelty And Significance:** 3
**Recommendation:** 8

**Clarity, Quality, Novelty And Reproducibility:**

Overall, the writing is clear and, considering the appendix provided, likely provides sufficient information for reproducibility. I am not fully up to date on the latest literature in the sub-area, but from what I am aware, I believe the authors' approach is novel and that their related work is thorough.

**Strength And Weaknesses:**

Strengths:
The authors tackle an important problem, and present a nice approach with a relatively simple intuition. I think their proxy for identifying OOD items is clever.
The mathematical derivation is thorough (though I have not checked its details)
Their evaluation is convincing

Weaknesses:
I'd like to see an explicit evaluation of their proxy for identifying OOD items
It would be nice to see more of a qualitative evaluation of which type of items their approach improved performance over, and which it didn't. Is the improvement randomly distributed over all OOD items, or are there qualitative properties of some items that dictated how well it would work?



**Summary Of The Paper:**

The authors introduce D-BAT, a diversity-inducing regularizer for training ensembles of diverse predictors. They derive D-BAT mathematically, and evaluate it on several datasets to demonstrate that the induced diversity can help to (i) tackle shortcut learning, and (ii) improve uncertainty estimation and transferability.

**Summary Of The Review:**

I think this paper provides an interesting and novel approach that makes progress on an important problem, and documents it well. I recommend acceptance.

---

> ### Author Response · Authors · 2022-11-18
> **Response to reviewer nyu8**
>
> Thank you for the time you took to review our paper, and for your appreciation of this work.
>
> > I'd like to see an explicit evaluation of their proxy for identifying OOD items It would be nice to see more of a qualitative evaluation of which type of items their approach improved performance over, and which it didn't.
>
> Our analysis of uncertainty estimation and OOD detection in Sec. 4.2 attempts at exactly this. In Fig. 5, we interpolate and extrapolate between a digit of 0 and 1. As is expected, the DBAT score is lowest (i.e. most certain) at the real images, and progressively increases (becomes more uncertain) as we interpolate or extrapolate. Similarly, in Fig. 6 we see that DBAT scores trained on CIFAR 10 appropriately disagree more (and are hence less confident) on OOD Cifar100 data. We agree that a more in-depth analysis of D-BAT’s OOD detection capabilities is interesting and plan to investigate this in future work.

---

### Official Review · Reviewer_Xnkg · 2022-10-25

**Confidence:** 4
**Correctness:** 3
**Technical Novelty And Significance:** 3
**Empirical Novelty And Significance:** 3
**Recommendation:** 8

**Clarity, Quality, Novelty And Reproducibility:**

## Clarity (W1)

The paper is overall well-written. The one part I found confusing is the theory on page 5. You say
>  Thus, the error on OOD data might be very high

and then proceed to _upper-bound_ the error on OOD data: $L_\text{ood}(h_\text{ERM}, h_\text{ood}) \le \max_{h \in H^*_t} L_\text{ood} (h, h_\text{ood})$. Then, you say that ideally we would like to minimize the right hand side. It is not clear what exactly you mean, minimize what with respect to what. I would guess that you mean that we wish to find a solution $h$ that minimizes the OOD loss $L_\text{ood} (h, h_\text{ood})$.

Then, you say you derive a proxy: $L_\text{ood}(h_1, h_\text{ood}) = \max_{h_2 \in H^* \cap H^*_\text{ood}} L_\text{ood}(h_1, h2)$. I don't see why this equality would hold. Maybe it should be $\le$ and not $=$?

Finally you arrive at the bound
$L_\text{ood}(h_1, h_\text{ood}) \le \max_{h_2 \in H^*} L_{\text{ood}} (h_1, h_2)$, which makes sense. However, how does this bound suggest that
> we want to pick $h_2$ to minimize our training data, but otherwise maximally disagree with $h_1$ on the OOD data

? I would agree that this point is suggested if our goal was to upper-bound the OOD loss of $h_1$, but it is not clear why this would be a good strategy for finding a solution $h_2$ with low OOD loss. I don't think the theory suggests it.

## Novelty (W4)

The authors mention the work [1] which has similar ideas, but trains diverse classifiers with a fixed feature extractor. I think it would be nice to include an explicit comparison with this method for a subset of your experiments. [2] suggests that fixed feature extractors may be sufficient, at least for datasets such as waterbirds and [3] suggests the same for Office-Home.

More generally, it would be nice to verify that diverse features are actually learned, rather than diverse classification heads on similar features.

## Experiments (W2, W3)

The improvements on Waterbirds and Office-Home are fairly small, especially given that ERM is the only baseline. In particular, on Waterbirds the standard metric in the spurious correlation literature is the worst group accuracy, which for strong methods is above 90%. For the authors, the average accuracy on the test set is 88%, suggesting that the worst group accuracy is probably much lower.

The only natural dataset where the method shows really impressive results is Camelyon17, where the authors seem to achieve state-of-the-art. My understanding is that the baseline numbers are reported from  the WILDS leaderboard (https://wilds.stanford.edu/leaderboard/#with-unlabeled-data-2). One difference I notice is that the methods in the leaderboard use a DenseNet-121 model, while you are using a ResNet-50 model.

**Q1**: Generally, did you use the official WILDS scripts to evaluate your method? If you switch the architecture to a DenseNet-121, would your method count as a "standard submission" by the WILDS leaderboard definitions (https://wilds.stanford.edu/submit/)?

**Q2**: Another question I have is why are the ensemble results missing for both ERM and D-BAT on all the synthetic datasets?


**Strength And Weaknesses:**

## Strengths

**S1**: The paper is interesting, and overall well-written. The presentation is logical
**S2**: The proposed method is intuitive
**S3**: The experiments show a big improvement in performance on the Camelyon17 dataset compared to strong baselines
**S4**: Experiments on uncertainty estimation show nice results.

## Weaknesses

**W1**: The theory is somewhat confusing.
**W2**: The improvements over the ERM are quite small on Waterbirds and Office-Home.
**W3**: The Camelyon17 experiments appear to use a somewhat different setup compared to the baselines (although I don't think it provides a significant advantage to D-BAT).
**W4**: Would be nice to have a direct comparison with [1] in the experiments.

I expand on the weaknesses below.

**Summary Of The Paper:**

The paper proposes D-BAT, a method for training diverse predictions by maximizing disagreement between the models on an OOD dataset. The authors provide a theoretical motivation for the method, and demonstrate improved performance on some OOD generalization and uncertainty estimation benchmarks.

**Summary Of The Review:**

Overall, this is an interesting paper which proposes an interesting method. I hope that the authors can address my questions above during the rebuttal.


## References:

[1] Evading the Simplicity Bias: Training a Diverse Set of Models Discovers Solutions with Superior OOD Generalization
Damien Teney, Ehsan Abbasnejad, Simon Lucey, Anton van den Hengel

[2] Last Layer Re-Training is Sufficient for Robustness to Spurious Correlations
Polina Kirichenko, Pavel Izmailov, Andrew Gordon Wilson

[3] Domain-Adjusted Regression or: ERM May Already Learn Features Sufficient for Out-of-Distribution Generalization
Elan Rosenfeld, Pradeep Ravikumar, Andrej Ristesk

---

> ### Author Response · Authors · 2022-11-14
> **Response to reviewer Xnkg**
>
> We thank the reviewer for his time reading our work and for the helpful suggestions.
>
> > The authors mention the work [1] which has similar ideas, but trains diverse classifiers with a fixed feature extractor. I think it would be nice to include an explicit comparison with this method for a subset of your experiments.
>
> In App F.1 we provide a comparison between [1] and D-BAT on one of our synthetic datasets. While  [1] has the merit of not relying on an OOD distribution, we show the method requires many heads to learn diverse features. Also note that D-BAT can be combined with a fixed pre-trained feature extractor. This is equivalent to applying D-BAT on the input space, but keeping the weights of the pre-trained network fixed.
>
> > [2] suggests that fixed feature extractors may be sufficient, at least for datasets such as waterbirds and [3] suggests the same for Office-Home. More generally, it would be nice to verify that diverse features are actually learned, rather than diverse classification heads on similar features.
>
> Our work looks at neural networks in a black box manner, whereas [2,3] identify that the feature extractors and the classification layer behave differently. An interesting implication of [2,3] is that, in most benign settings (where the correlation between spurious and non-spurious features is low), D-BAT only needs to be applied on the last few layers. This, along with an explanation for their empirical observation, would be excellent future directions to investigate.
>
> > The one part I found confusing is the theory on page 5.
> > I don't see why this equality would hold. Maybe it should be ≤ and not = ?
>
> The reasoning goes like this (with slightly altered notations: $L\\equiv\\mathcal{L}$, $D\\equiv\\mathcal{D}$ and $H\\equiv\\mathcal{H}$):
> * We have $L_{D_{ood}}(h_1,h_{ood}) = L_{D_{ood}}(h_1,h_2),  \\forall h_2 \\in H_{ood}^\\star$ as all $h_2 \\in H_{ood}^\\star$ are functionally equivalent on $D_{ood}$.
> * Therefore, in particular: $L_{D_{ood}}(h_1,h_{ood}) = \\max_{h_2 \\in H_{ood}^\\star \cap H_t^\\star} L_{D_{ood}}(h_1,h_2)$
> * And finally: $\\max_{h_2 \\in H_{ood}^\\star \\cap H_t^\\star} L_{D_{ood}}(h_1,h_2) \\leq \\max_{h_2 \\in H_t^\\star} L_{D_{ood}}(h_1,h_2)$
>
> We modified the submission to clarify the first step.
>
> > I would agree that this point is suggested if our goal was to upper-bound the OOD loss of h1, but it is not clear why this would be a good strategy for finding a solution h2  with low OOD loss. I don't think the theory suggests it.
>
> The argument makes more sense when considering the ensemble. By picking $h_2$ optimizing the bound, we minimize the expected loss between members of the ensemble $\\{h_1,h_2\\}$ and the worst case $h_{ood}$, i.e. we minimize $\\min_{h \\in \\{h_1, h_2\\}}\\max_{h' \\in H_t^\\star} L_{D_{ood}}(h,h')$. The intuition is clearer in Fig.3.
>
> > Generally, did you use the official WILDS scripts to evaluate your method?
>
> That is a good observation, while we are using our own code to evaluate our models (see the `get_acc` function in the `src/utils.py` code). In the case of Camelyon17, we verified that both the official WILDS script and ours are giving exactly the same scores. Our reported test accuracy matches exactly the average accuracy (`acc_avg`) returned by the `wilds_dataset.eval` function which is also used to compute the scores reported in Sagawa et al. (2022). Our Camelyon17 results are therefore comparable.
>
> > why are the ensemble results missing for both ERM and D-BAT on all the synthetic datasets?
>
> We did not include them so as not to overload the table and because we felt the results are trivial - the synthetic datasets were explicitly constructed such that ERM would always pick the simpler feature and D-BAT would always find a diverse ensemble. We can include these detailed results in the Appendix.

---

> > ### Comment · Reviewer_Xnkg · 2022-11-17
> > **Thank you for the rebuttal**
> >
> > Thank you for your response, clarifications, and updates.

---

### Decision · Program_Chairs · 2023-01-20

**Decision:**

Accept: notable-top-5%

**Justification For Why Not Higher Score:**

N/A

**Justification For Why Not Lower Score:**

Reviewers all gave high scores (8) and raised no serious complaints, rather heaped praise on the paper. I think this is an important topic. The solution seems well motivated AND general, which is a rarity in transfer learning research, in my opinion. I'm commending oral. It would not be a travesty for this to be a spotlight, depending on the competitiveness of the field this year.

**Metareview: Summary, Strengths And Weaknesses:**

This paper proposes to improve transfer learning by learning a diverse collection of predictors. The new idea is that one can find a usefully diverse collection of predictors by regularization that enforces DISagreement on out-of-distribution data (and agreement on in-distribution data). Subsequently, the evaluate either a uniform mixture ensemble of this diverse collection or they use labelled OOD data to perform model selection. (The diversity regularization requires only unlabelled data.)

This is an interesting idea and empirical results (on 6 small'ish data sets) suggests that the approach has merit. Reviewers generally agreed that the paper was well written, well motivated, and well researched. The appendices contain comparisons and additional discussions. From my vantage point, I would have liked to have seen more extensive experimentation, but I think there's more than enough here to warrant acceptance.

**Note From Pc:**

if the above contains the word "oral" or "spotlight" please see: "oral" presentation means -> notable-top-5% and "spotlight" means -> notable-top-25%. As stated in our emails, we are disassociating presentation type from AC recommendations

**Summary Of Ac-Reviewer Meeting:**

N/A